# Waste to Energy: Solid Fuel Production from Biogas Plant Digestate and Sewage Sludge by Torrefaction-Process Kinetics, Fuel Properties, and Energy Balance

**Kacper Świechowski [1],\*** , **Martyna Hnat [1]** , **Paweł Stępień [1]** , **Sylwia Stegenta-Dąbrowska [1]** , **Szymon Kugler [2]** , **Jacek A. Koziel [3]** **and Andrzej Białowiec [1,3]**

[1] Faculty of Life Sciences and Technology, Institute of Agricultural Engineering, Wrocław University of Environmental and Life Sciences, 37/41 Chełmońskiego Str., 51-630 Wrocław, Poland; hnat.martyna@gmail.com (M.H.); pawel.stepien@upwr.edu.pl (P.S.); sylwia.stegenta@upwr.edu.pl (S.S.-D.); andrzej.bialowiec@upwr.edu.pl or andrzejb@iastate.edu (A.B.)

[2] Faculty of Chemical Technology and Engineering, Polymer Institute, West Pomeranian University of Technology, 10 Pułaskiego Str., 70-322 Szczecin, Poland; szymon.kugler@zut.edu.pl

[3] Department of Agricultural and Biosystems Engineering, Iowa State University, Ames, IA 50011, USA; koziel@iastate.edu

\* Correspondence: kacper.swiechowski@upwr.edu.pl

**Abstract:** Sustainable solutions are needed to manage increased energy demand and waste generation. Renewable energy production from abundant sewage sludge (SS) and digestate (D) from biogas is feasible. Concerns about feedstock contamination (heavy metals, pharmaceuticals, antibiotics, and antibiotic-resistant bacteria) in SS and D limits the use (e.g., agricultural) of these carbon-rich resources. Low temperature thermal conversion that results in carbonized solid fuel (CSF) has been proposed as sustainable waste utilization. The aim of the research was to investigate the feasibility of CSF production from SS and D via torrefaction. The CSF was produced at 200~300 °C (interval of 20 °C) for 20~60 min (interval 20 min). The torrefaction kinetics and CSF fuel properties were determined. Next, the differential scanning calorimetry (DSC) and thermogravimetric analysis (TGA) of SS and D torrefaction were used to build models of energy demand for torrefaction. Finally, the evaluation of the energy balance of CSF production from SS and D was completed. The results showed that torrefaction improved the D-derived CSF's higher heating value (*HHV*) up to 11% ($p < 0.05$), whereas no significant *HHV* changes for SS were observed. The torrefied D had the highest *HHV* of 20 MJ·kg$^{-1}$ under 300 °C and 30 min, (the curve fitted value from the measured time periods) compared to *HHV* = 18 MJ·kg$^{-1}$ for unprocessed D. The torrefied SS had the highest *HHV* = 14.8 MJ·kg$^{-1}$ under 200 °C and 20 min, compared to *HHV* 14.6 MJ·kg$^{-1}$ for raw SS. An unwanted result of the torrefaction was an increase in ash content in CSF, up to 40% and 22% for SS and D, respectively. The developed model showed that the torrefaction of dry SS and D could be energetically self-sufficient. Generating CSF with the highest *HHV* requires raw feedstock containing ~15.4 and 45.9 MJ·kg$^{-1}$ for SS and D, respectively (assuming that part of feedstock is a source of energy for the process). The results suggest that there is a potential to convert biogas D to CSF to provide renewable fuel for, e.g., plants currently fed/co-fed with municipal solid waste.

**Keywords:** renewable energy; sewage sludge; biogas digestate; waste to energy; waste to carbon; circular economy; sustainability; carbonized solid fuel

# 1. Introduction

## *1.1. Abundant Waste Resources for Solid Fuel Production*

The energy use per capita grew from 1.3 to 1.9 Mg of oil equivalents in 1971–2014 [1]. The global energy demand is expected to grow by about 27% worldwide from 2017 to 2040 [2]. The increase in energy needs and consumption has an impact on the environment [3,4]. There is a need to refine technologies for clean, abundant, and renewable energy for sustainable development.

Waste production increases with development. In general, developed economies produce more waste (mainly plastic), whereas in emerging economies, citizens generate less waste that nonetheless has a high content of high organic biodegradables. Regardless of the development stage, sewage sludge (SS) is abundantly produced worldwide as a byproduct of wastewater treatment. For example, Poland generated over 584,000 Mg d.m. (dry mass) of SS and over 9,300,000 Mg d.m. was produced in the whole EU in 2017 [4].

Biogas digestate (D) is another abundant source of carbon-rich waste that is a byproduct of renewable energy production. Two billion $m^3$ (bcm) of biogas are produced annually in the EU, representing ~0.42% of the total natural gas consumed (470 bcm). It is estimated that the amount of biogas produced in 2050 will be 36~98 bcm [5,6]. Such a significant increase will be associated with a challenge to find sustainable waste management of the produced D. Currently, the European biogas market is concentrated in Germany, with more than half of all European biogas plants located there [5]. Thus, new plants are likely to be built throughout the EU, which will create a market for D utilization.

The EU generates ~180 mln Mg of D per year. Approximately 120 mln Mg is produced from agricultural substrates, ~46 mln Mg from mixed municipal solid waste, 7 mln Mg from separated biowaste, and the remainder from SS and other agro/food industry by-products [7]. These Ds are directly used as fertilizer [7].

## *1.2. Waste Management Policies Create an Opportunity for Sustainable Reuse of SS and D*

The EU has introduced policies regarding the increase of the share of renewable energy in total energy consumption and to waste management. For example, Directive 2009/28/EC promoted renewable energy and assumed its growth to be at least 20% of the total energy consumption in 2020 [8]. Though it is known that some EU countries did not achieve this goal, the EU established a new target for 2030 that assumes at least a 32% share for renewable energy [9].

Transition to the circular economy has been promoted. Directive 2009/28/EC [8] laid down measures to prevent or reduce the adverse impacts of the generation and management of waste [8]. The directive established a waste hierarchy (article 4) that relegated conventional incineration and landfilling while promoting prevention and re-use [8]. This transition creates an opportunity to find sustainable re-uses of SS that is current landfilled or incinerated.

The technologies for thermal waste treatment need to adjust to the shift from incineration and high-energy input to the medium- and low-energy input of non-recyclable residual waste. It has been agreed that incineration plants will continue to be an important element of waste management and a proper mix should be maintained when it comes to the waste-to-energy capacity for the treatment of non-recyclable waste. This is critical to avoid potential economic losses or the creation of infrastructural barriers to the achievement of higher recycling rates [10]. However, the unintended effect of increased recycling will be less fuel for incineration plants and a lower fuel quality. This is because the biggest calorific fraction (e.g., plastic) will be sorted out from the waste stream.

The European Committee for Standardization (CEN) established quality standards for solid recovered fuels (SRFs) to address the high variability and heterogeneity of waste streams and to simplify the market of waste conversion to energy. The EN 15359:2012 divides fuels produced from waste into five classes based on their low heating value (*LHV*), chlorine, and mercury content. The *LHV* for the first through fifth classes are ≥25, ≥20, ≥15, ≥10, and ≥3 MJ·kg$^{-1}$, respectively [11]. The chlorine content is responsible for the temperature in which SRF can be incinerated, whereas mercury is

a main environmental concern. Other SRF parameters include corrosion and deposits to build up compounds [12]. Unprocessed waste, like SRF, contain some biological shares (home for harmful mold, fungus, and microorganism, virus, etc.), as well as small plastics particles; for this reason, they can be sources of health problems for people having contact with these materials [13].

### 1.3. Valorization of Waste via Torrefaction

Large quantities of SS and D are still used in agriculture [14,15]. However, a large fraction of SS and D waste streams cannot be used for fertilization due to its contamination (e.g., heavy metals, pharmaceuticals, antibiotics, and antibiotic-resistant bacteria [16–20]). Contaminated SS needs to be stabilized and then landfilled or incinerated [14], i.e., approaches that are being phased out in the EU. Similarly, some Ds from municipal biogas plants do not meet fertilizer standards. Biological hazards, dust, and lower calorific values of waste from sorting plants can be overcome by the thermal conversion of SRF to carbonized solid fuel (CSF), followed by CSF densification via pelletization. Thermal treatment (e.g., via torrefaction) eliminates biological hazards and increases energy density, and the pelletization further improves the energy densification and reduction of volatile organic compound (VOC) emissions from CSF up to 86% [21,22].

Thus, there is an opportunity for SS valorization to high-quality fuel via torrefaction. Torrefaction is a thermal treatment known as 'mild pyrolysis', 'roasting', or 'high-temperature drying'. Torrefaction is known to upgrade the fuel characteristics of biomass [23]. Torrefaction can also overcome the disadvantages of raw biomass, such as high moisture content, degradation and decay, odor, pathogens, and low energy density. The torrefaction process increases hydrophobicity and reduces grinding energy demand [24]. Torrefaction is achieved via the relatively slow heating of biomass at 200~300 °C in a no or limited oxygen environment [23].

This research aimed to investigate the feasibility of producing CSF from dry SS and D and completing initial techno-economic analyses for CSF utilization in cement and power plants. In this work, dried SS and D were torrefied and then compared to other alternative CSFs. This research addresses the goals of (1) an increasing share of renewables, (2) providing additional options for solid fuel for power plants in the future, and (3) managing the growing volume of organic waste produced by energy recovery.

The torrefaction experiment and process modeling were done for dry SS and D, excluding the energy needed for a drying process. The torrefaction of dry materials instead of materials with natural moisture contents was chosen for several reasons: (i) The initial moisture content of SS and D is very high <90% and its direct torrefaction could be biased (i.e., SS could be incinerated autothermal when its moisture content is under 50% [25]; (ii) SS is already dried to avoid landfill costs so that it can be incinerated or used to produce solid fertilizer [25] in larger wastewater treatment plants (in selected EU countries); (iii) moreover, technologies for water removal from SS and D by mechanical or thermal treatment are available, including solar drying [26] and/or waste heat from other processes [27]— for example, waste heat from biogas incineration in combined heat and power (CHP) units can be used for D drying; finally (iv), it is assumed that the model developed for dry mass will be easier to use, i.e., by recalculating for site-specific SS and D conditions (taking into account the initial moisture and the energy cost of its removal).

## 2. Materials and Methods

The experiment setup is presented in Figure 1. A detailed description is below. First, samples of digestate and sewage sludge were collected from industrial plants. Next, the samples were dried and ground. Then, parts of the samples were processed to CSF sample generation. The dried samples of raw SS and D were tested by thermogravimetric analysis (TGA) and differential scanning calorimetry (DSC) analyses. In parallel, the dried raw SS and D and CSF samples were tested by proximate and process analyses. After that, data analysis was conducted. Finally, as a result of data analysis, empirical models of CSF fuel features, torrefaction kinetics, and energy balance were obtained.

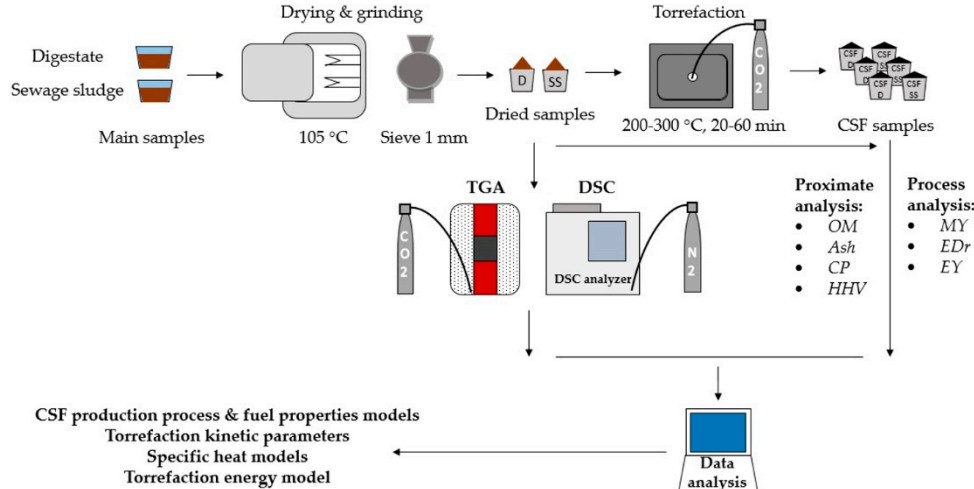

**Figure 1.** Experiment setup to convert sewage sludge and digestate to carbonized solid fuel (CSF) via torrefaction. The resulting CSF was analyzed for inputs to techno-economic analyses. D = biogas plant digestate; SS = sewage sludge. TGA = thermogravimetric analyses. DSC = differential scanning calorimeter analysis; *OM* = organic matter; *CP* = combustible parts; *HHV* = high heating value; *MY* = mass yield; *EDr* = energy densification ratio; and *EY* = energy yield.

## 2.1. Feedstock

### 2.1.1. Sewage Sludge

SS was collected at the 140,000 $m^3 \cdot d^{-1}$ wastewater treatment plant (WWTP) (Janówek, MPWiK S.A., Wrocław, Poland). The SS was a by-product of mechanical and biological wastewater treatment, with chemical additives for phosphorus removal. The 20 kg SS sample was collected from the secondary settling tank before the anaerobic digestion. Then, the sample was dried at 105 °C in a laboratory dryer (WAMED, model KBC-65W, Warsaw, Poland). Next, the dry SS was ground through a 1 mm screen with a laboratory knife mill (Testchem, model LMN-100, Pszów, Poland) and then stored before testing at −15 °C.

### 2.1.2. Digestate from the Biogas Plant

D originated from the 1 $MW_{el}$ commercial biogas plant (Bio-Wat Sp. Z o. o., Świdnica, Poland). The biogas plant used the following feedstocks: a biodegradable fraction of municipal solid waste (34%), maize silage (30%), sugar beet pulp (30%), and yeast cake (6%). The 20 kg D sample was collected from the post-fermentation chamber. Next, the sample was dried, ground, and stored in identical conditions to that of SS.

## 2.2. CSF Production Method and Process Analysis

The CSF was produced in accordance with the previously described methodology [28]. A muffle furnace (Snol 8.1/1100, Utena, Lithuania) was used. $CO_2$ was delivered to the center of the furnace at ~2.5 $dm^3 \cdot min^{-1}$ to facilitate an inert atmosphere. Furnace setpoint temperatures of 200~300 °C (with 20 °C intervals) and 20~60 min (20 min intervals) residence times were used. The (10 ± 0.5 g) dry SS and D feedstock samples were heated in inert conditions from room temperature (20 °C) with a heating rate of 50 °C·min$^{-1}$ to the setpoint. After the torrefaction process, CSF samples were removed from the muffle furnace when the interior temperature was lower than 200 °C. The approximate times of cooling from 300, 280, 260, 240, and 220–200 °C were ~38, 33, 29, 23, and 13.5 min, respectively. A process temperature vs. process time for 300 °C setpoint is presented in Figure 2. The mass of the sample before and after torrefaction was determined to calculate the mass loss and yield. The mass was measured within 0.1 g of accuracy.

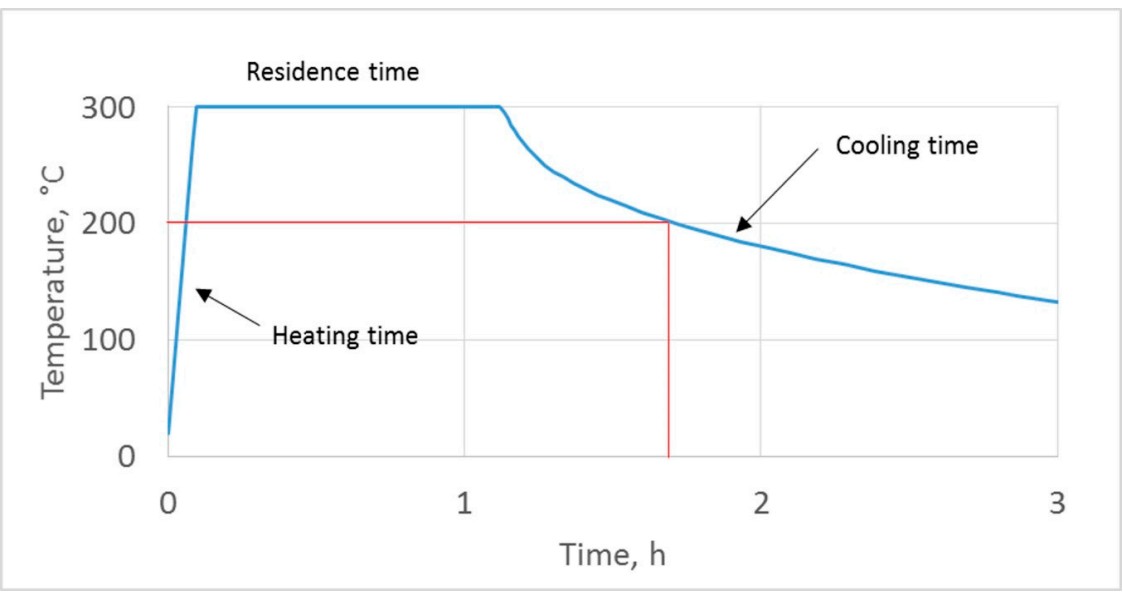

**Figure 2.** An example of temperature patterns during the torrefaction of sewage sludge and digestate.

The mass yield, energy densification ratio, and energy yield of CSF were determined based on Equations (1), (2), and (3) [29], respectively.

$$MY = m_b/m_a \cdot 100 \tag{1}$$

where *MY* is the mass yield (%), $m_a$ is the mass of raw material before torrefaction (kg), and $m_b$ is the mass of CSF after torrefaction (kg).

$$EDr = HHV_b/HHV_a \tag{2}$$

where *EDr* is the energy densification ratio, $HHV_b$ is the high heating value of CSF (MJ·kg$^{-1}$), and $HHV_a$ is the high heating value of raw material (MJ·kg$^{-1}$).

$$EY = MY \cdot EDr \tag{3}$$

where *EY* is the energy yield (%), *MY* is the mass yield (%), and *EDr* is the energy densification ratio.

### 2.3. Proximate Analysis of SS and D and their CSF

The physical–chemical properties of dry SS and D and CSF were tested in three replicates for:

- Organic matter (*OM*) content, a.k.a. a loss on ignition (*LOI*), using the method described elsewhere [30].
- Combustible part (*CP*) and ash content (*ash*) [31].
- High heating value (*HHV*) [32].

The (Snol 8.1/1100, Utena, Lithuania) furnace was used for *OM*, *CP*, and ash determination. The C200 calorimeter (IKA®Werke GmbH, Staufen, Germany) was used for *HHV* determination.

#### The Properties of Raw Feedstock

The *OM* content for dry SS and D was 61.9% and 86.6%, respectively. The ash content was 36.3% and 12.4% for SS and D (d.m.), respectively. The *CP* in dried SS and D were 63.7% and 87.6%, respectively. The *HHV* (14.6 MJ·kg$^{-1}$) of dried SS was lower than for dried D (18.1 MJ·kg$^{-1}$).

### 2.4. Thermogravimetric Analysis (TGA) of Raw Sewage Sludge and Digestate

The thermogravimetric analysis was performed in isothermal and non-isothermal conditions. First, isothermal conditions were used in order to determine the kinetics parameters (*k*—constant;

reaction rate; $E_a$—activation energy; and $A$—pre-exponential factor) of the torrefaction process. Next, non-isothermal conditions were used for tracking the thermal degradation from 50 to 850 °C.

The determination of kinetic parameters was completed in accordance with the previous methodology and reactor set-up [33]. Setpoint torrefaction temperatures and 1 h heating time in inert $CO_2$ ~10 $dm^3 \cdot h^{-1}$ flowrates were used for mass losses based on the initial mass of the dry sample (2.25 g) in three replicates. Next, the mass losses for each torrefaction temperature setpoints were used to determine constant reaction rates $k$. The first-order model was used (Equation (4)):

$$m_s = m_o \cdot e^{(-k \cdot t)} \tag{4}$$

where $m_s$ is mass at time t (g), $m_o$ is initial mass (g), $k$ is the reaction rate constant ($s^{-1}$), and $t$ is time (s).

The full methodology of kinetic parameters determination ($k$—constant reaction rate; $Ea$—activation energy; and A—pre-exponential factor) was presented in a previous work [34].

TGA in non-isothermal conditions was carried out at a heating rate of 10.8 $°C \cdot min^{-1}$. Dry SS and D samples were placed in a tubular reactor and then heated to 850 °C, and they were kept there for 2 min.

The analysis of kinetic parameters and thermal degradation was done by means of the stand-mounted tubular furnace (Czylok, RST 40x200/100, Jastrzębie-Zdrój, Poland).

Data from non-isothermal TGA were subjected to mathematical manipulation in accordance with the following description. Raw TGA data were smoothed by using the locally weighted scatterplot smoothing (LOWESS) method [35] with *Span* (0–1) = 0.1. Next, based on the smoothed TGA curve, a derivative thermogravimetric curve (DTG) was created with the Savitzky–Golay smooth method (polynomial order = 2 and points of window = 20) [35]. The OriginPro 2017 software (OriginLab, Northampton, MA, USA) was used for data analysis.

## 2.5. Differential Scanning Calorimetry (DSC) of Raw Material

The DSC of SS and D was carried out in $N_2$ (3 $dm^3 \cdot h^{-1}$) atmosphere using a differential scanning calorimeter (TA Instruments, DSC Q2500, New Castle, DE, USA). The dry SS and D sample (~6 mg) was placed into the aluminum crucible, placed in the calorimeter, and heated from 20 to 500 °C (at 10 $°C \cdot min^{-1}$) in n = 1 replicate.

## 2.6. Modeling of Torrefaction Process and CSF Fuel Properties

Polynomial models of the influence of torrefaction temperature and process (residence) time on the CSF parameters (*MY*, *EDr*, *EY*, *OM* content, *CP* content, ash content, and *HHV*) were developed. Models were based on measured data from the torrefaction and CSF properties for a particular torrefaction temperature and time using a similar modeling approach described in our previous work [36]. The general model is presented by Equation (5). Each model had one intercept ($a_1$) and six regression coefficients ($a_2$—$a_7$) (a confidence interval of 95% was assumed). Regression coefficients for which the $p$-value was <0.05 were assumed to be statistically significant. Correlation ($R$) and determination coefficients ($R^2$) were determined for each model.

$$f(T,t) = a_1 + a_2 \cdot T + a_3 \cdot T^2 + a_4 \cdot t + a_5 \cdot t^2 + a_6 \cdot T \cdot t + a_7 \cdot T^2 \cdot t^2 \tag{5}$$

where $f(T,t)$ is the variable ($T$, $t$, and combinations) being analyzed, $a_1$ is the intercept, $a_2$—$a_7$ are the regression coefficients, $T$ is the torrefaction process temperature (°C), and $t$ is torrefaction process time (min).

The standardized regression coefficients $\beta$ for each regression coefficients ($a_2$—$a_7$) were standardized based on Equation (6). The $\beta$ coefficient determines how much its own standard deviations will change the dependent variable $Y$ if the independent variable is changed by one (its own) standard deviation [36].

$$\beta = a_n \cdot SD_{Xi} / SD_{Yi} \tag{6}$$

where $\beta$ is the standardized regression coefficient, $a_n$ is the estimated regression coefficient, $SD_{Xi}$ is the standard deviation of the independent variable $x$, $xi$ represents the values of subsequent independent variables, $SD_{Yi}$ is the standard deviation of the dependent variable $y$, and $yi$ represents the values of subsequent dependent variables.

### 2.7. Energy Balance for Torrefaction

An energy balance of the torrefaction process was needed to determine if the process could be self-sustaining. The calculations were aimed to determine the energy needed to generate 1 g of CSF. The energy balance assumed:

- No heat losses of the reactor.
- The heat needed to dry SS and D were not included (due to site-specific variability in the feedstock and drying methods).
- All energy contained in torrgas was used to provide energy to the torrefaction process.
- The energy contained in torrgas was estimated based on Equation (8).

The energy balance model is presented in Figure 3. Material for torrefaction is given as the *HHV* of raw material multiplied by its mass needed ($x$) to obtain 1 g of CSF after the process. The $x$ is calculated as:

$$x = 1/MY \cdot 100 \tag{7}$$

where $x$ is a multiplier for an additional raw material mass to compensate for mass loss during torrefaction, $MY$ is the mass yield of the torrefaction process (values based on the model, in %), and 100 is the value to remove the % unit from the equation.

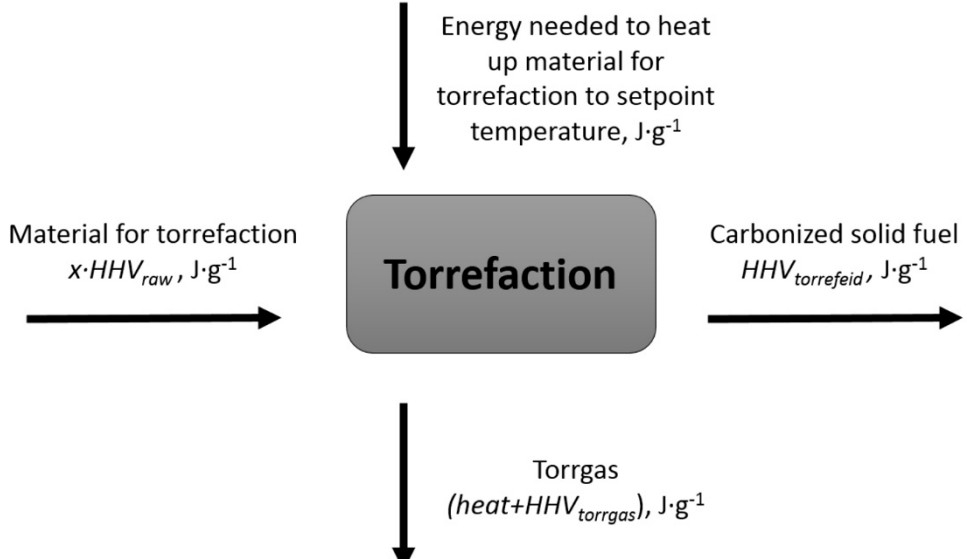

**Figure 3.** Energy balance of torrefaction to produce CSF (energy/mass).

The energy consumption of the torrefaction was estimated similarly to the model developed by Stępień et al. [37]. The model calculates energy needed to heat material to the setpoint temperature of torrefaction and uses the TGA and DSC analyses. In this research, the energy required to heat SS and D from 20 °C (room temperature) to 200, 220, 240, 260, 280, and 300 °C was estimated and then increased by multiplying it by $x$ value to determine the energy needed to produce 1 g of CSF. The energy contained in a torrgas was calculated as Equation (8). Equation (8) was based on the assumption that total energy contained in torrgas (heat (energy contained in gas temperature) and chemical (energy contained in torrgas composition)) was a sum of external energy delivered to heat up material and energy contained in released volatiles minus the energy that remained in CSF. In reality,

the total energy potential of torrgas is lower than the calculated one due to the heat loss when CSF is removed from the reactor to the cooling stage (energy from the CSF cooling process was omitted for ease of calculations).

$$E_{torrgas} = E_{heat\ up} + E_{raw} - E_{CSF} \tag{8}$$

where $E_{torrgas}$ is the energy contained in torrgas (J·g$^{-1}$), $E_{heat\ up}$ is the energy needed to heat dry SS or D to setpoint temperature to produce 1 g of CSF (J·g$^{-1}$), $E_{raw}$ is the energy contained in raw material (dry SS or D) before torrefaction used to obtain 1 g of CSF (J·g$^{-1}$), and $E_{CSF}$ is the energy contained in 1 g of the obtained CSF (J·g$^{-1}$).

If the energy contained in torrgas was higher than the energy needed to heat materials SS or D to the setpoint temperature, it was assumed that the process of CSF generation was self-sufficient. The energy contained in 1 g of obtained CSF was calculated as *HHV* based on the *HHV* results.

## 3. Results

Raw data from the tests described in Sections 2.2–2.5 are presented in the Supplementary Materials. The results from the particular tests were tabulated on five excel sheets. The first sheet "Read Me" is a guide about how to find data. The sheet "Torrefaction Process" contains the results of process mass yield, energy densification ratio, and energy yield. The sheet "Proximate Analysis" contains results of moisture content, organic matter content, combustible part content, ash content, and high heating value of the tested materials. Next, the sheets named "TGA (Isothermal Condition)" and "TGA (Non-Isothermal Condition)" contain results from the thermogravimetric analysis. The last sheet "DSC" contains results from differential scanning calorimetry.

### 3.1. The Effect of Torrefaction Temperature and Time on CSF Properties

The mass yields (*MY*) for SS and D torrefaction decreased with an increase of process temperature (Figure 4). This trend was more apparent for the D than for SS. At 300 °C and 60 min torrefaction, *MY* was ~80% and 40% for SS and D, respectively. The highest *MY* values were obtained for CSFs generated at the lowest temperature (200 °C). For both models, all regression coefficients were statistically significant ($p < 0.05$) (Table A1), and determination coefficients ($R^2$) were >0.83, which indicates a reasonable fit to the experimental data. For the SS model, the most important coefficient was $a_6$ ($\beta = -6.27$), whereas, in the D model, it was $a_6$ ($\beta = -4.34$). The sum of standardized $\beta$ coefficients ($a_2$—$a_7$) for these models was $-0.4$ and $-0.52$, respectively, for SS and D (Table A1), which means that generally, the *MY* value was decreasing with the increase of torrefaction temperature and process time.

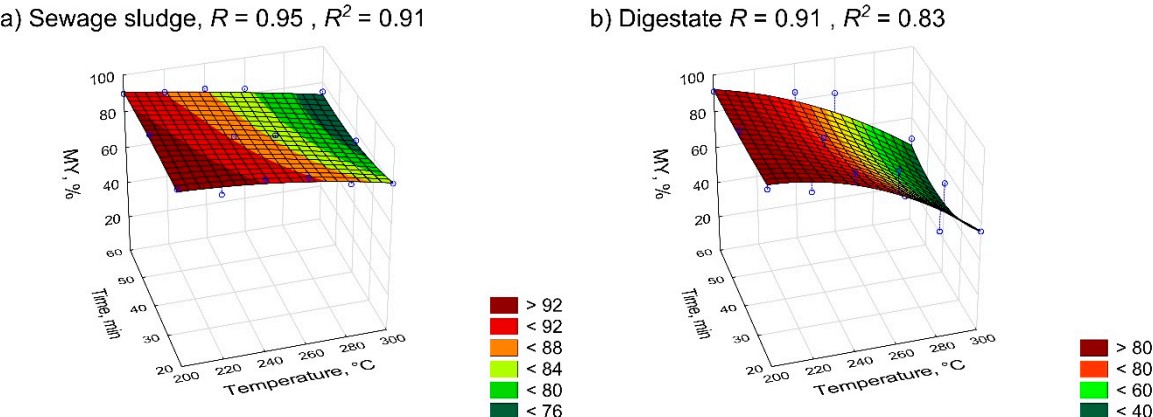

**Figure 4.** The influence of torrefaction temperature and residence time on the mass yield of CSF from (**a**) sewage sludge and (**b**) digestate. *R*—correlation coefficient; $R^2$—determination coefficient.

The *EDr* in CSF generated from SS decreased with an increase of process temperature, whereas it increased for D (Figure 5). CSFs from SS produced at 200 °C had an *EDr* of ~1.01, while CSFs

generated at 300 °C had an *EDr* of ~0.85. For CSFs generated from D, *EDr* values were ~1.01–1.10. It appears that CSF production from D was promoted by short residence time up to ~40 min and high torrefaction temperature (280–300 °C). For SS, time did not have an impact on *EDr* and was promoted at low temperatures (200–240 °C) (Figure 5). For both models, all regression coefficients were statistically significant ($p < 0.05$) (Table A2), while the $R^2$ was 0.85 and 0.68 for SS and D, respectively. The most important coefficient was $a_3$ ($\beta = -4.20$), whereas, in the D model, it was $a_6$ ($\beta = -12.91$) (Table A2). The sum of standardized $\beta$ coefficients ($a_2$–$a_7$) was −0.56 and −0.49, respectively, for SS and D (Table A2). This means that generally, the *EDr* value decreased with the increase of torrefaction temperature and process time. It is somewhat surprising in the case of D where *EDr* increased, but this increase was not consistent across the studied range; the *EDr* decrease was apparent for torrefaction longer than ~40 min and higher than ~260 °C (Figure 5).

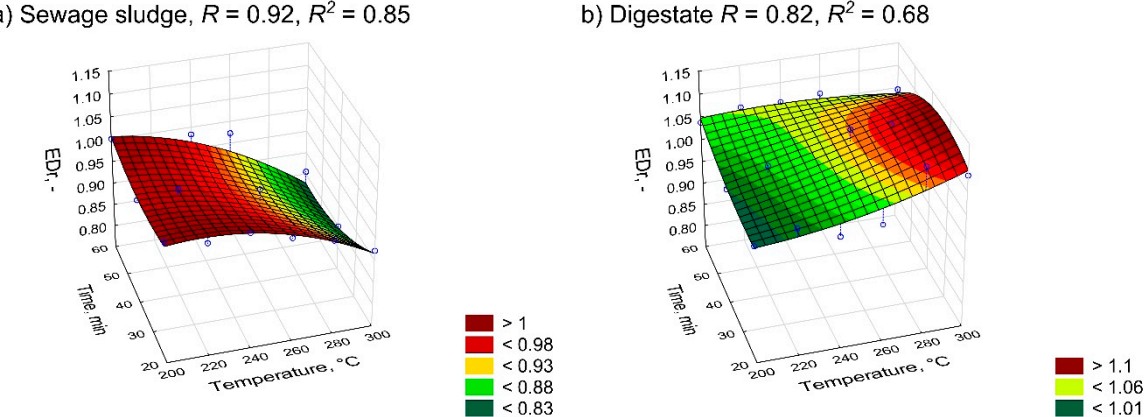

**Figure 5.** The influence of torrefaction temperature and residence time on the energy densification ratio of CSF from (**a**) sewage sludge and (**b**) digestate. *R*—correlation coefficient; $R^2$—determination coefficient.

The *EY* for CSF decreased with an increase in process temperature for both SS and D. The time had a lower impact on *EY* compared to the temperature (Figure 6). For SS, the *EY* decreased from ~100 to ~60%, whereas for D, it decreased from ~100 to ~45%. All regression coefficients were statistically significant ($p < 0.05$) (Table A3), with $R^2 = 0.9$ and 0.83 for SS and D, respectively. The most important coefficients were $a_6$ ($\beta = -4.55$) for SS and $a_3$ ($\beta = -3.79$) for D (Table A3). The sum of the standardized $\beta$ coefficients ($a_2$–$a_7$) for these models was −0.55 and −0.59, respectively, for SS and D (Table A3), which means that generally, *EY* decreased with the increase of torrefaction temperature and process time.

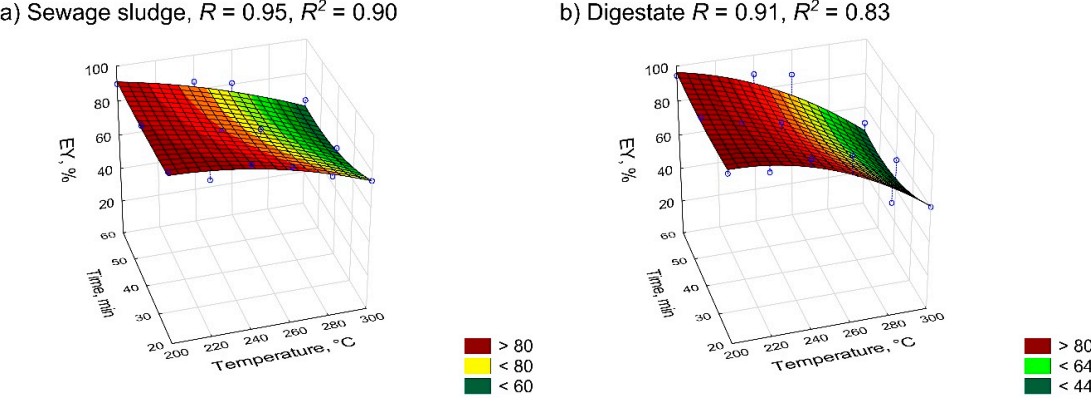

**Figure 6.** The influence of torrefaction temperature and residence time on the energy yield of CSF from (**a**) sewage sludge and (**b**) digestate. *R*—correlation coefficient; $R^2$—determination coefficient.

### 3.2. Result of Proximate Analysis of CSF

The *OM* content in CSFs decreased with an increase in temperature. CSF from SS was characterized by a lower *OM* (~57~47%) compared with D-derived CSF (~87%~75%) (Figure 7). The time and temperature had a significant impact ($p < 0.05$) on decreasing *OM*. Statistical differences between particular measurements are given in the Tables A8 and A9. There were no differences in *OM* ($p < 0.05$) for CSFs generated from SS in a range from 200 °C (20~60 min) to 220 (20~40 min) (Table A8). In the case of D, more differences between particular process ranges ($p < 0.05$) were found (Table A9). The $R^2$ values for SS and D were 0.89 and 0.83, respectively. All regression coefficients were statistically significant ($p < 0.05$) (Table A4). The most important coefficient was $a_6$ ($\beta = -5.15$) for SS and $a_3$ ($\beta = -2.83$) for D (Table A4). The sum of the standardized $\beta$ coefficients ($a_2$–$a_7$) for these models was $-0.42$ and $-0.77$, respectively (Table A4). This means that, generally, the *OM* value decreased with the increase of torrefaction temperature and process time. The sum of the $\beta$ coefficients for D was lower than for SS; the total loss in the organic matter was greater for D (Table A4).

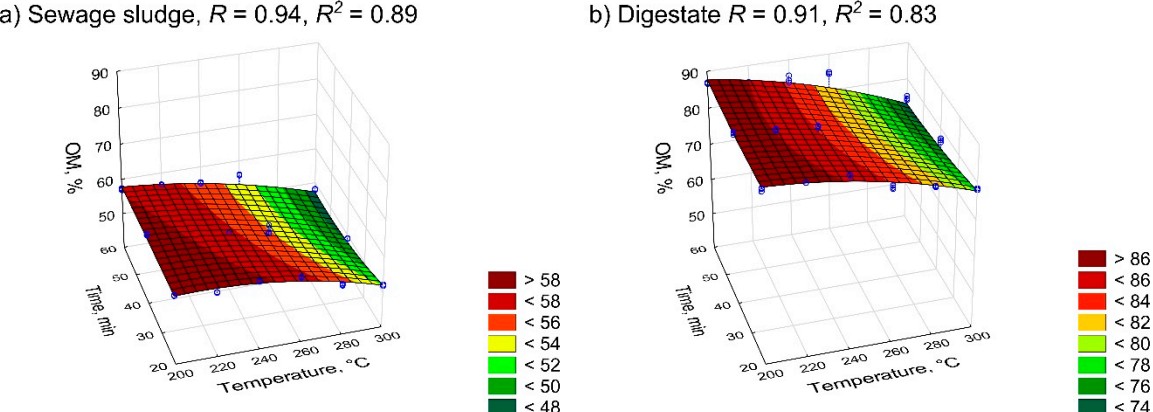

**Figure 7.** The influence of torrefaction temperature and residence time on the organic matter content in CSF from (**a**) sewage sludge and (**b**) digestate. *R*—correlation coefficient; $R^2$—determination coefficient.

The ash content ranged from ~40% to ~48% and from ~12% to ~24% for SS and D, respectively. Torrefaction increased the ash content in CSF from both SS and D, and it was significant with the increase in temperature and residence time ($p < 0.05$) (Figure 8). The statistical differences between ash content for particular conditions are given in Tables A10 and A11. There were no differences in ash content ($p < 0.05$) (Table A10) in CSFs from SS produced at 200 °C for 40~60 min, and up to 240 °C for 20~60 min. All regression coefficients were statistically significant ($p < 0.05$) (Table A5), and the models' $R^2$ values were 0.88 and 0.82 for SS and D, respectively. The most important coefficient for the SS model was $a_6$ ($\beta = 5.10$), whereas it was $a_4$ ($\beta = 2.47$) for the D model (Table A5). The sum of the standardized $\beta$ coefficients ($a_2$–$a_7$) for these models was 0.42 and 1.22, respectively, for SS and D (Table A5). This means that the ash content generally increased with the increase of torrefaction temperature and process time. The sum of the $\beta$ coefficients was higher for D than for SS; the CSF production from D was characterized by a faster increase in ash content (relative to the initial ash content of the raw material).

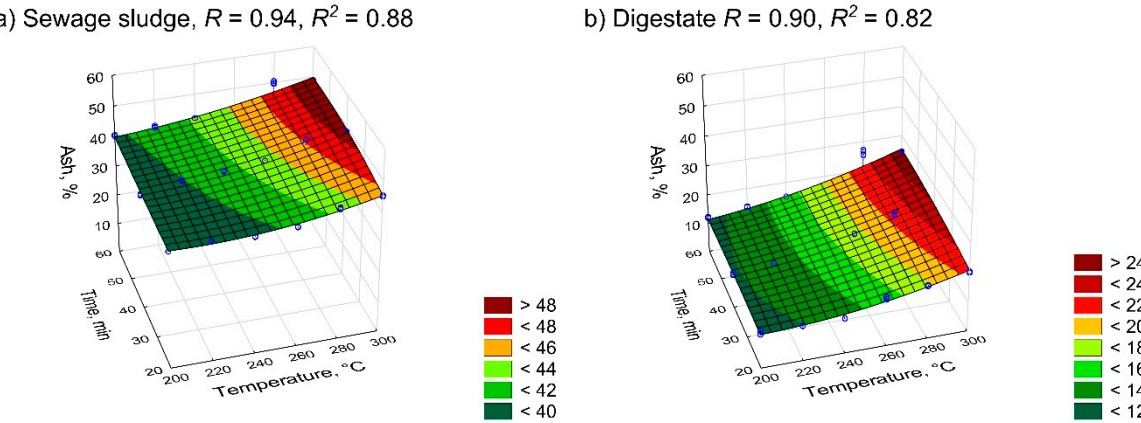

**Figure 8.** The influence of torrefaction temperature and residence time on ash content in CSF from (**a**) sewage sludge and (**b**) digestate. *R*—correlation coefficient; $R^2$—determination coefficient.

The content of *CP* had an opposite trend to ash content. *CP* decreased from ~60% to ~52% and from ~88% to ~76% for SS and D, respectively (Figure 9). There were no differences in *CP* in CSFs from SS produced at 200 °C for 40~60 min up to 240 °C for 20~60 min ($p < 0.05$) (Table A12), similar to the trend observed for the ash content. The statistical differences between the *CP* of D-derived CSFs were varied (Table A13). All regression coefficients were statistically significant ($p < 0.05$) (Table A6). Both models had high $R^2$ values of 0.88 and 0.82, for SS and D, respectively. The most important coefficient for the SS model was $a_6$ ($\beta = -5.10$), whereas it was $a_4$ ($\beta = -2.47$) for the D model (Table A6). The sum of the standardized $\beta$ coefficients ($a_2$–$a_7$) for these models was −0.42 and −1.22, respectively, for SS and D (Table A6). This trend was the opposite one to observed for ash content.

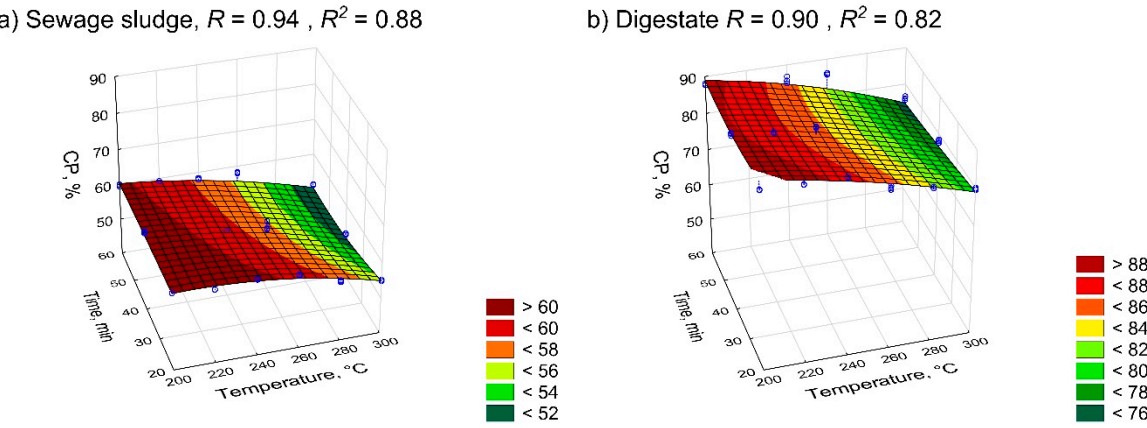

**Figure 9.** The influence of torrefaction temperature and residence time on combustible parts in CSF from (**a**) sewage sludge and (**b**) digestate. *R*—correlation coefficient; $R^2$—determination coefficient.

The torrefaction for SS resulted in a decrease of *HHV* from ~14 to ~13 MJ·kg$^{-1}$ with an increase in residence time and process temperature (Figure 10a). However, an increase in *HHV* with temperature was observed for the D where the *HHV* increased up to 40 min. The longer torrefaction of D past 40 min caused the *HHV* to decrease again. The highest value of *HHV* for D-derived CSF was ~20 MJ·kg$^{-1}$ at 300 °C and 40 min (Figure 10b). There were no statistical differences in *HHV* ($p < 0.05$) for D-derived CSFs produced from 200 °C for 40~60 min up to 280 °C for 20 min (Table A15). The statistical differences between the *HHV* of SS-derived CSFs are presented in Table A14. All regression coefficients were statistically significant ($p < 0.05$) (Table A7), and the $R^2$ of D was only 0.52; it was 0.81 for SS. The most important coefficient for the SS model was $a_6$ ($\beta = -4.09$), whereas it was $a_6$ ($\beta = 11.87$) for the D model (Table A7). The sum of standardized $\beta$ coefficients ($a_2$–$a_7$) was −0.55 and −0.46, for SS and D,

respectively (Table A7). The trends observed here were similar to *EDr*, namely, despite the increase of the *HHV* from a certain point, it began to decrease, allowing for process optimization.

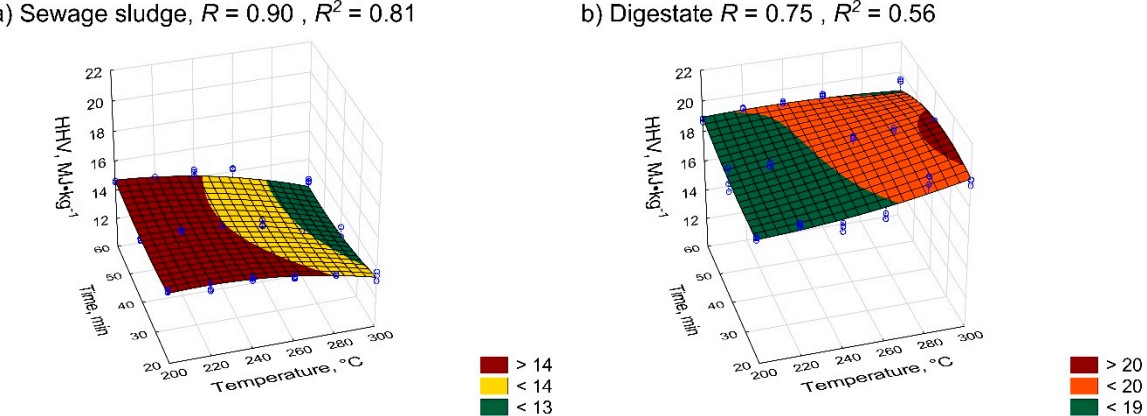

a) Sewage sludge, $R$ = 0.90 , $R^2$ = 0.81          b) Digestate $R$ = 0.75 , $R^2$ = 0.56

**Figure 10.** The influence of torrefaction temperature and residence time on the high heating value of CSF from (**a**) sewage sludge and (**b**) digestate. *R*—correlation coefficient; $R^2$—determination coefficient.

### 3.3. The Thermogravimetric Analysis

The reaction rates (*k*) constants for the first-order equation were calculated based on mass losses during torrefaction for each process temperatures (Figures 11 and 12 and Table 1). Next, an Arrhenius plot was created from *k* values, and then linear models were created (Figure 13), from which $E_a$ and *A* values were calculated. The determination coefficient for SS was higher than for D ($R^2$ = 0.99 vs. $R^2$ = 0.90, respectively) (Figure 13). The *k* for 200~280 °C was higher for SS ($k = 8.71 \times 10^{-6} \sim 2.99 \times 10^{-5}$), whereas at 300 °C, the *k* value of D was greater ($k = 4.60 \times 10^{-5}$) (Table 1). The $E_a$ and *A* parameters ranged from 46,700 to 52,230 and from 0.75 to 1.95, respectively (Table 1).

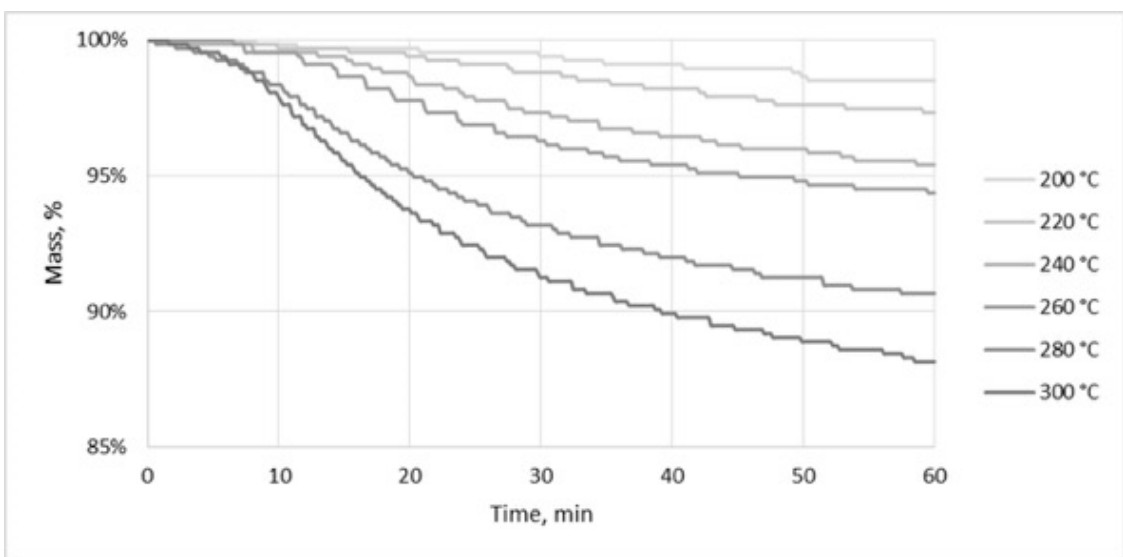

**Figure 11.** TGA of sewage sludge at torrefaction temperatures.

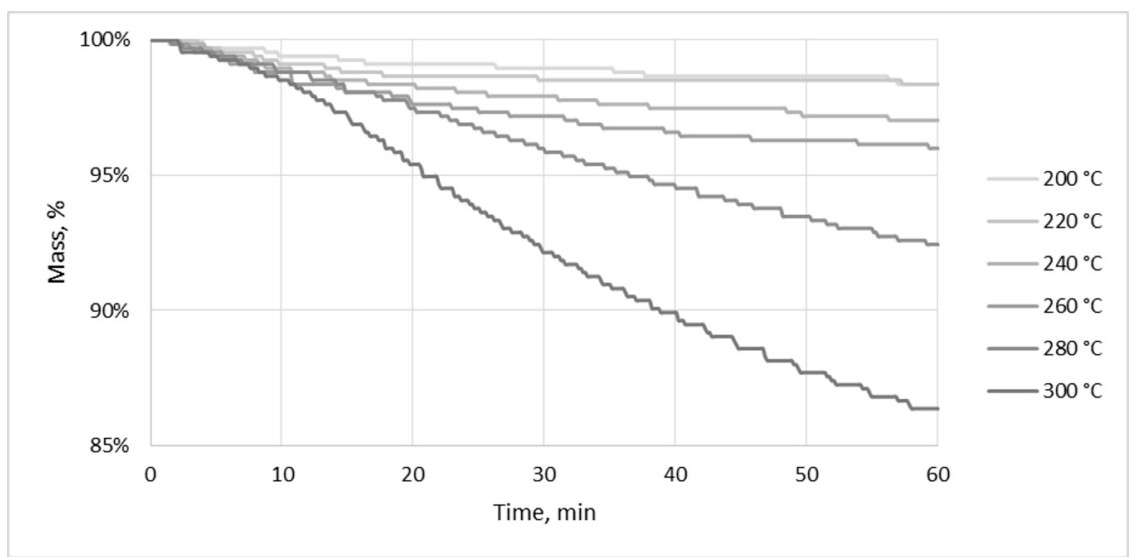

**Figure 12.** TGA of digestate at torrefaction temperatures.

**Table 1.** Summary of kinetic parameters of the torrefaction process.

| Material | T, °C | T, K | k, s$^{-1}$ | E$_a$, J·mol$^{-1}$ | A, s$^{-1}$ |
|---|---|---|---|---|---|
| Sewage sludge | 200 | 473 | $4.73 \times 10^{-6}$ | 46,700 | 0.75 |
| | 220 | 493 | $8.71 \times 10^{-6}$ | | |
| | 240 | 513 | $1.52 \times 10^{-5}$ | | |
| | 260 | 533 | $1.90 \times 10^{-5}$ | | |
| | 280 | 553 | $2.99 \times 10^{-5}$ | | |
| | 300 | 573 | $3.85 \times 10^{-5}$ | | |
| Digestate | 200 | 473 | $4.91 \times 10^{-6}$ | 52,230 | 1.95 |
| | 220 | 493 | $4.46 \times 10^{-6}$ | | |
| | 240 | 513 | $7.79 \times 10^{-6}$ | | |
| | 260 | 533 | $1.09 \times 10^{-5}$ | | |
| | 280 | 553 | $2.34 \times 10^{-5}$ | | |
| | 300 | 573 | $4.60 \times 10^{-5}$ | | |

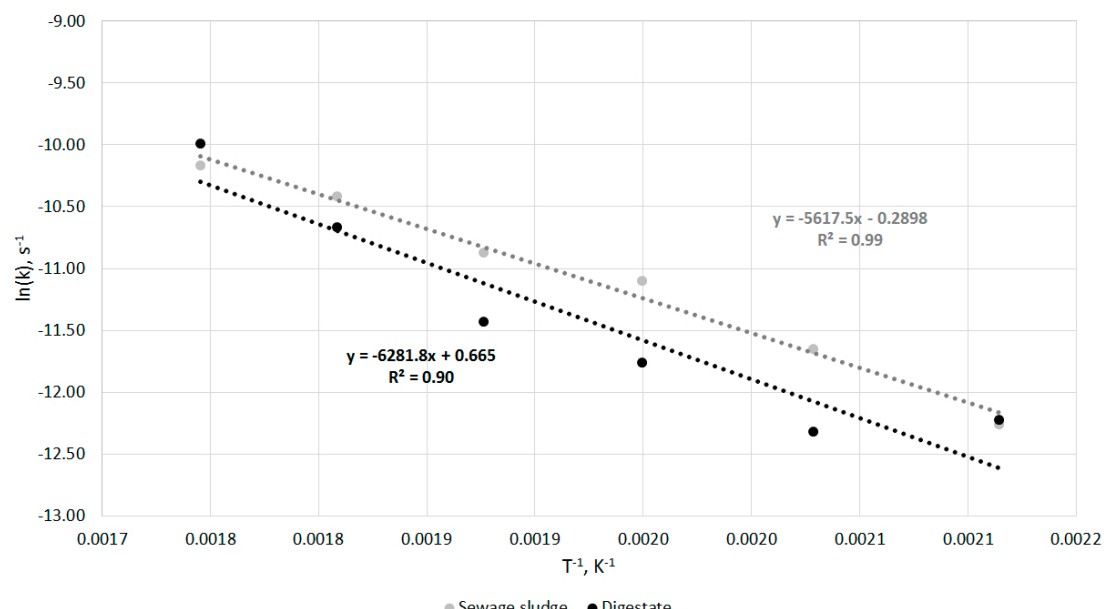

**Figure 13.** Arrhenius plot for sewage sludge and digestate.

Figure 14 presents a thermal decomposition in an inert condition under non-isothermal conditions for SS and D samples heated from 50 to 850 °C. The decomposition of SS started at ~200–240 °C, whereas the decomposition of D started at 260–270 °C. After ~450 °C, the thermal decomposition of D sped up compared to SS, and at the end (850 °C), D had an average weight loss of ~63%, whereas SS had one of ~50% (Figure 14). The principal decomposition of the D started at ~350 °C and ended at ~550 °C, with a maximum decomposition peak at ~475 °C (DTG = 0.5%). For SS, a principal thermal decomposition started earlier at ~300 °C and ended at ~700 °C, with a maximum decomposition peak at ~500 °C (DTG = 0.2%).

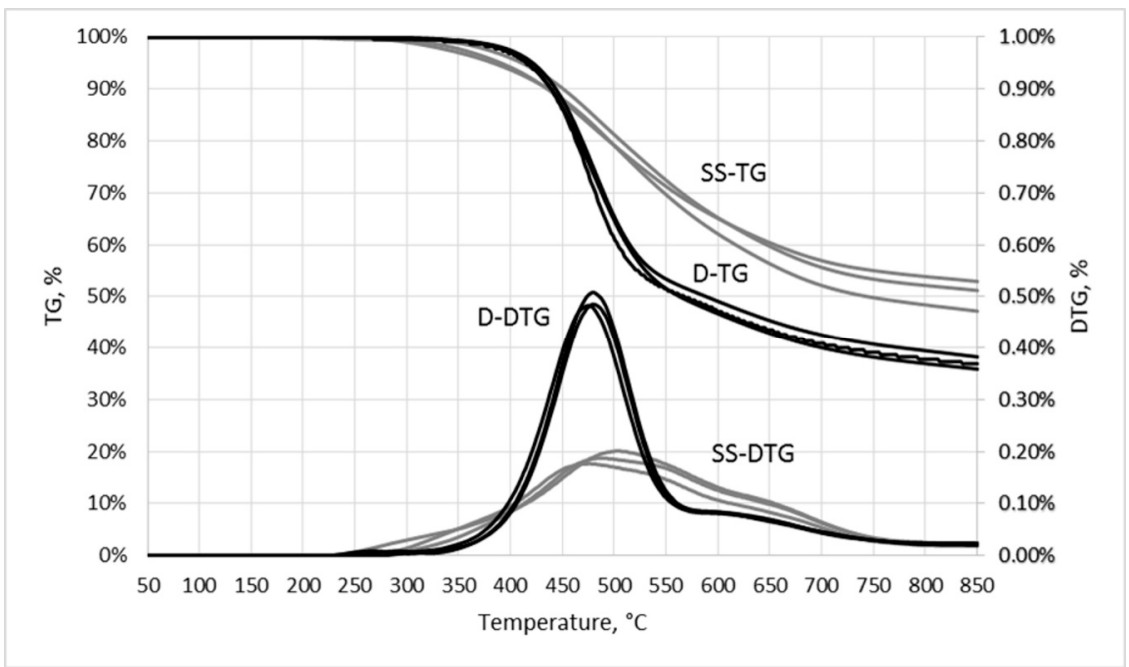

**Figure 14.** The thermogravimetric characteristic of sewage sludge (SS) and digestate (D) heated from 50 °C to 850 °C.

### 3.4. The Differential Scanning Calorimetry

Differential scanning calorimetry analysis revealed heat flow characteristics and energy needs to heat SS and D from 20 °C to 500 °C with a heating rate of 10 °C·min$^{-1}$ in a nitrogen atmosphere. One endoenergetic transformation occurred for SS; the transformation started at 72 °C and ended at 170 °C (Figure 15). During transformation, two peaks occurred—first at 102 °C and second at 155 °C. The total energy needed for this transformation was 21.53 J·g$^{-1}$. After transformation, the energy needs for heating SS started to decrease. The decrease of heat flow with an increase of temperature from 170 to 500 °C was almost linear (Figure 15).

In the case of D, two transformations occurred. The first one was an endothermic transformation. It started at 36 °C and ended at 168 °C. The second transformation was exothermic. It started at 285 °C and ended at 351 °C, with a maximum peak at 327 °C. The total energy needed for the endothermic reaction was 115.19 J·g$^{-1}$, whereas the exothermic one emitted 39.84 J·g$^{-1}$ (Figure 16).

The energy demand for heating SS and D to the setpoint of torrefaction was estimated based on results from the TGA analysis (Figure 14) and DSC analysis (Figures 15 and 16). Since the estimations were based on dried SS and D, the energy needed for water removal was not included. The energy demand estimation was completed based on the protocol proposed by Stępień et al. [37]. Then, the energy needed to produce 1 g of CSF was estimated as the "energy needed to heat 1 g of raw material" (Table 2) multiplied by *x* (Equation (7)). The results showed that heating 1 g of SS from 20 to 200–300 °C required more energy (449–643 J·g$^{-1}$) compared to the energy needed for heating of 1 g of digestate (381~492 J·g$^{-1}$) to the same torrefaction setpoint (Table 2). Due to the mass loss occurring

during the process, the *MY* decreased, and, therefore, the *x* value (Equation (7)) increased from 1.05 to 1.37 and 1.02 to 2.30 for SS and D, respectively (Table 2). The energy needed to produce 1 g of CSF increased with torrefaction temperature and time in the case of both SS and D. The decreasing trend of energy contained in CSF produced from SS in higher torrefaction temperatures and times conditions for SS was observed, whereas for D, the trend was opposite.

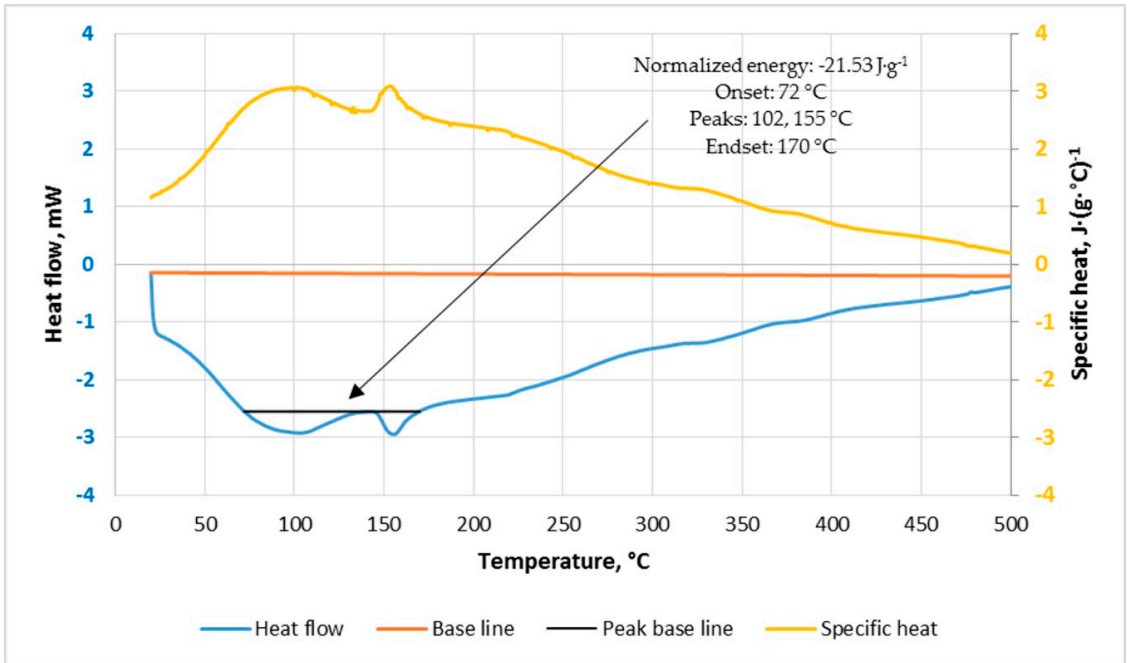

**Figure 15.** DSC analysis of sewage sludge.

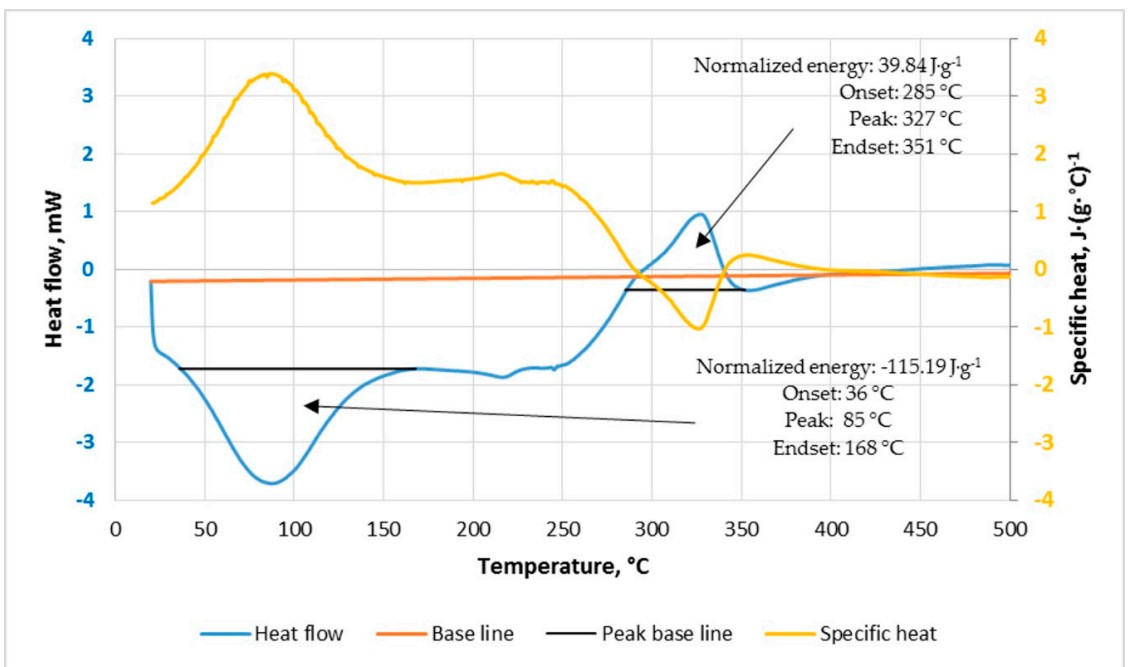

**Figure 16.** DSC analysis of digestate.

**Table 2.** Results of torrefaction energy balance.

| Feedstock | Torrefaction temperature, °C | Torrefaction residence time, min | Energy needed to heat up 1 g of raw material, J·g$^{-1}$ | MY, % | x, - | Energy needed to produce 1 g of CSF, J·g$^{-1}$ | Energy contained in raw material used to produce 1 g of CSF, J·g$^{-1}$ | Energy contained in 1 g of CSF, J·g$^{-1}$ | Energy contained in torrgas (heat and HHV$_{torrgas}$), J·g$^{-1}$ |
|---|---|---|---|---|---|---|---|---|---|
| Sewage sludge | 200 | 20 | 449 | 94.94 | 1.05 | 473 | 15,368 | 14,692 | 1150 |
| | | 40 | 449 | 93.54 | 1.07 | 480 | 15,597 | 14,414 | 1663 |
| | | 60 | 449 | 90.42 | 1.11 | 497 | 16,135 | 14,668 | 1964 |
| | 220 | 20 | 496 | 93.86 | 1.07 | 529 | 15,544 | 14,758 | 1315 |
| | | 40 | 496 | 90.71 | 1.10 | 547 | 16,084 | 14,322 | 2309 |
| | | 60 | 496 | 87.35 | 1.14 | 568 | 16,704 | 14,456 | 2816 |
| | 240 | 20 | 540 | 92.00 | 1.09 | 587 | 15,859 | 14,655 | 1791 |
| | | 40 | 540 | 87.30 | 1.15 | 619 | 16,712 | 14,066 | 3265 |
| | | 60 | 540 | 84.06 | 1.19 | 642 | 17,358 | 14,088 | 3912 |
| | 260 | 20 | 579 | 89.34 | 1.12 | 648 | 16,330 | 14,383 | 2596 |
| | | 40 | 579 | 83.32 | 1.20 | 695 | 17,512 | 13,646 | 4560 |
| | | 60 | 579 | 80.55 | 1.24 | 719 | 18,113 | 13,565 | 5267 |
| | 280 | 20 | 613 | 85.90 | 1.16 | 714 | 16,986 | 13,941 | 3759 |
| | | 40 | 613 | 78.76 | 1.27 | 779 | 18,525 | 13,062 | 6241 |
| | | 60 | 613 | 76.83 | 1.30 | 798 | 18,989 | 12,887 | 6900 |
| | 300 | 20 | 643 | 81.66 | 1.22 | 787 | 17,868 | 13,330 | 5325 |
| | | 40 | 643 | 73.62 | 1.36 | 873 | 19,817 | 12,314 | 8376 |
| | | 60 | 643 | 72.90 | 1.37 | 882 | 20,013 | 12,054 | 8840 |
| Digestate | 200 | 20 | 381 | 98.14 | 1.02 | 388 | 18,432 | 18,122 | 698 |
| | | 40 | 381 | 94.18 | 1.06 | 405 | 19,209 | 18,210 | 1404 |
| | | 60 | 381 | 91.70 | 1.09 | 415 | 19,728 | 18,822 | 1322 |
| | 220 | 20 | 413 | 96.11 | 1.04 | 430 | 18,823 | 18,269 | 984 |
| | | 40 | 413 | 88.95 | 1.12 | 465 | 20,336 | 18,521 | 2280 |
| | | 60 | 413 | 86.34 | 1.16 | 479 | 20,952 | 18,900 | 2530 |
| | 240 | 20 | 444 | 90.76 | 1.10 | 489 | 19,931 | 18,505 | 1915 |
| | | 40 | 444 | 80.86 | 1.24 | 549 | 22,371 | 18,864 | 4056 |
| | | 60 | 444 | 78.85 | 1.27 | 563 | 22,944 | 18,915 | 4591 |
| | 260 | 20 | 472 | 82.12 | 1.22 | 575 | 22,029 | 18,828 | 3776 |
| | | 40 | 472 | 69.91 | 1.43 | 676 | 25,877 | 19,238 | 7314 |
| | | 60 | 472 | 69.21 | 1.44 | 683 | 26,138 | 18,868 | 7953 |
| | 280 | 20 | 490 | 70.17 | 1.43 | 699 | 25,779 | 19,240 | 7238 |
| | | 40 | 490 | 56.09 | 1.78 | 875 | 32,254 | 19,644 | 13,485 |
| | | 60 | 490 | 57.43 | 1.74 | 854 | 31,498 | 18,757 | 13,594 |
| | 300 | 20 | 492 | 54.92 | 1.82 | 895 | 32,936 | 19,740 | 14,091 |
| | | 40 | 492 | 39.40 | 2.54 | 1248 | 45,918 | 20,082 | 27,085 |
| | | 60 | 492 | 43.52 | 2.30 | 1130 | 41,571 | 18,584 | 24,117 |

## 4. Discussion

*4.1. The Impact of Torrefaction Technological Parameters on the Efficiency of the Process and Fuel Properties*

The *MY* of SS and D showed a decreasing trend during torrefaction. *MY* decreased with the increase of the process temperature. The *MY* of SS decreased to ~80%, whereas for D, it decreased up to ~40% at 300 °C (Figure 4). The decreasing trend for both tested materials was also shown in the case of *EY*. Despite twice differences for *MY*, the *EY* differences were smaller, i.e., the tested SS and D contained 60% and 50% of their initial energy content, respectively, for the maximum torrefaction temperature of 300 °C (Figure 6). The 10% difference in *EY* resulted from differences in the *EDr* (Figure 5). For SS, the value of *EDr* decreased, whereas for D, *EDr* increased with the increase of temperature. Differences in *EDr* were likely a result of differences in *OM* and the composition and thermal reactivity of SS and D. However, the *MY* of SS torrefaction was comparable to other studies (Table 3). Torrefaction is feasible for CSF production for additional types of abundant waste feedstock (refuse-derived fuel, sawdust, pruned biomass, walnut shells, spent mushroom compost, and elephant dung) [34,38–42].

Pulka et al. [38] torrefied a SS, originating from different wastewater treatment plants, by means of a tubular furnace at temperatures 200–300 °C for 1 h with a resulting *MY* of 90~80%. The *MY* of the torrefaction process of D can be compared to lignocellulose materials such as Oxytree pruning biomass or sawdust (*MY* 92~55% and 94~33%, respectively) (Table 3). The lower *MY* of D and lignocellulose materials resulted from much lower ash content in raw materials. There were over 30% of ash in the biomass waste (Table 3), which resulted in a decrease of *OM* content due to its decomposition during torrefaction (Table 3). The decreasing trend of *EDr* with temperature and time for SS (Figure 5) was also confirmed by Pulka et al. [38], where *EDr* was 0.96–0.29 ($T$ = 200~300 °C, $t$ = 1 h). Compared to the results in this study, *EDr* was ~1.0–0.83 for the same conditions (Figure 5). D-derived CSF showed an uptrend for *EDr* for 20 and 40 min, while for 60 min, *EDr* (1.05) was stable regardless of the process temperature (Figure 5). The uptrend of *EDr* resulted from the torrefied material energy densification. A similar uptrend was also visible in other materials such as Oxytree pruned biomass, reuse-derived fuel (RDF), and sawdust (Table 3).

The properties of the tested SS were *OM* = 61.9%, ash = 36.3%, *CP* = 63.7%, and *HHV* = 14.6 MJ·kg$^{-1}$. Similar values for SS-derived CSF were reported by Pulka et al. [38], where *OM*, ash, and *HHV* were 56.2%, 43.1%, and 13.5 MJ·kg$^{-1}$, respectively. For the tested D, proximate analyses showed *OM*, ash, *CP*, and *HHV* of 86.6%, 12.4%, 87.6%, and 18.1 MJ·kg$^{-1}$, respectively. The main outcome of the analysis was that SS had a higher ash content and, therefore, a lower *HHV* than D. In comparison to elephant dung (the product of methane fermentation in the elephant stomach) [34], the tested D from biogas plant had over five times less ash content and was comparable *HHV* (Table 3). It follows that the initial fermentation of substrates has a crucial influence on final product properties. In terms of energy content, SS and D were incomparable with typical energy biomass substrates, e.g., *Miscanthus x Giganteus*, *Rosa multiflora* (energetic rose), and *Salix viminalis* (willow) that have an *HHV* of 17.68, 17.54, and 17.5 MJ·kg$^{-1}$, respectively [43].

This study showed that *OM* (Figure 7) and *CP* (Figure 9) decreased with the increase of process temperature and time, whereas the ash content increased ($p < 0.05$) (Figure 8) (Tables A8–A13) for both SS and D. This effect was expected during biomass torrefaction and confirmed by other works. The organic compounds of biomass are degraded under high temperatures and are removed in the form of gas, whereas inorganic materials remain in biomass [36,38].

Ash acts as ballast; its higher concentration results in a decrease of energy fuel quality. In the tested SS, an initial high ash concentration (36.3%) contributed to a high ash concentration of ~40–50% in CSF (Figure 8). As a result of the devolatilization of *OM* and increased ash content, the *HHV* of SS-derived CSFs started to decrease with an increase of temperature and time (Figure 10). The *HHV* decreased from ~14 to ~13 MJ·kg$^{-1}$ (200–300 °C). Similar findings were obtained for elephant dung, where ash content increased by ~50–71% and *HHV* decreased from 11.4 to 6.5 MJ·kg$^{-1}$ (200~300 °C, 1 h) (Table 3). The reduction of energy content in solid residue after torrefaction was also reported by

Syguła et al. [40], where spent mushroom compost was torrefied. The calorific value of the torrefied biomass increased with process temperature up to 280 °C (13.8~17.8 MJ·kg$^{-1}$), whereas at 300 °C, *HHV* decreased to 14.3 MJ·kg$^{-1}$.

**Table 3.** Summarized results of the torrefaction process technological parameters for different waste materials (process time = 1 h).

| Material | Temperature, °C | MY, % | EDr | EY, % | OM, % | Ash, % | CP, % | HHV, MJ·kg$^{-1}$ | Reference |
|---|---|---|---|---|---|---|---|---|---|
| Sewage sludge | Raw | - | - | - | 56.2 | 43.1 | - | 13.5 | [38] |
| | 200 | 90 | 0.96 | 86 | 56.4 vm | 43.6 | - | 12.9 | |
| | 220 | 91 | 0.98 | 89 | 56.2 vm | 43.8 | - | 13.2 | |
| | 240 | 89 | 0.99 | 88 | 55.9 vm | 44.1 | - | 13.4 | |
| | 260 | 88 | 0.48 | 42 | 36.3 vm | 63.7 | - | 6.5 | |
| | 280 | 87 | 0.30 | 26 | 27.7 vm | 72.3 | - | 4.1 | |
| | 300 | 80 | 0.29 | 23 | 26.6 vm | 73.4 | - | 3.9 | |
| Elephant dung | Raw | - | - | - | 48.9 | 50.8 | 49.2 | 11.4 | [34] |
| | 200 | 96 | 1.14 | 109 | 57.4 | 42.5 | 57.5 | 13.0 | |
| | 220 | 90 | 1.12 | 102 | 60.2 | 39.8 | 60.2 | 12.8 | |
| | 240 | 89 | 0.83 | 74 | 49.8 | 50.1 | 49.9 | 9.5 | |
| | 260 | 90 | 0.91 | 82 | 44.8 | 55.1 | 44.9 | 10.3 | |
| | 280 | 63 | 0.81 | 52 | 28.3 | 71.5 | 28.5 | 7.5 | |
| | 300 | 73 | 0.86 | 63 | 28.7 | 71.3 | 28.7 | 6.5 | |
| Spent Mushroom Compost | Raw | - | - | - | 71.6 vm | 28.4 | - | 13.8 | [40] |
| | 200 | 97 | 1.07 | 103 | 69.7 vm | 30.3 | - | 14.4 | |
| | 220 | 99 | 1.18 | 116 | 76.7 vm | 23.3 | - | 15.9 | |
| | 240 | 96 | 1.10 | 105 | 71.5 vm | 28.5 | - | 14.8 | |
| | 260 | 95 | 1.16 | 110 | 69.8 vm | 30.2 | - | 15.5 | |
| | 280 | 93 | 1.33 | 123 | 68.2 vm | 31.8 | - | 17.8 | |
| | 300 | 90 | 1.06 | 95 | 57.4 vm | 42.6 | - | 14.3 | |
| Pruning Oxytree biomass | Raw | - | - | - | 90.2 | 8.1 | 91.9 | 18.3 | [41] |
| | 200 | 92 | 1.05 | 96 | 89.3 | 8.7 | 91.3 | 19.2 | |
| | 220 | 88 | 1.06 | 93 | 88.3 | 9.7 | 90.3 | 19.4 | |
| | 240 | 78 | 1.11 | 86 | 86.6 | 11.2 | 88.8 | 20.4 | |
| | 260 | 64 | 1.16 | 74 | 85.0 | 12.5 | 87.5 | 21.1 | |
| | 280 | 57 | 1.18 | 67 | 83.2 | 13.9 | 86.1 | 21.6 | |
| | 300 | 55 | 1.20 | 66 | 83.3 | 13.6 | 86.4 | 22.0 | |
| Walnut Shells | Raw | - | - | - | 81.4 vm | 0.6 | - | 19.6 | [42] |
| | 200 | 87 | 1.05 | 91 | 78.4 vm | 0.7 | - | 20.6 | |
| | 220 | 84 | 1.06 | 89 | 77.8 vm | 0.9 | - | 20.7 | |
| | 240 | 70 | 1.08 | 75 | 75.4 vm | 1.2 | - | 21.1 | |
| | 260 | 64 | 1.13 | 72 | 70.2 vm | 1.5 | - | 22.1 | |
| | 280 | 39 | 1.18 | 46 | 60.2 vm | 2.1 | - | 23.1 | |
| | 300 | 43 | 1.24 | 54 | 44.7 vm | 2.2 | - | 24.3 | |
| Refuse-Derived Fuel (RDF) | Raw | - | - | - | 76.0 vm | 14.3 | - | 26.9 | [39] |
| | 200 | 85 | 0.94 | 80 | 74.7 vm | 14.1 | - | 28.2 | |
| | 220 | 73 | 1.07 | 78 | 72.9 vm | 16.4 | - | 31.4 | |
| | 240 | 61 | 0.99 | 60 | 63.5 vm | 21.6 | - | 29.9 | |
| | 260 | 55 | 1.04 | 57 | 56.9 vm | 23.9 | - | 31.5 | |
| | 280 | 58 | 1.03 | 60 | 60.4 vm | 23.1 | - | 31.5 | |
| | 300 | 62 | 1.13 | 70 | 61.9 vm | 23.2 | - | 34.1 | |
| Sawdust | Raw | - | - | - | 77.6 vm | 0.5 | - | 19.6 | [39] |
| | 200 | 94 | 0.99 | 93 | 76.9 vm | 0.6 | - | 20.0 | |
| | 220 | 75 | 1.12 | 84 | 65.3 vm | 0.9 | - | 21.2 | |
| | 240 | 58 | 1.17 | 68 | 58.8 vm | 0.9 | - | 22.4 | |
| | 260 | 42 | 1.24 | 52 | 48.8 vm | 1.3 | - | 23.5 | |
| | 280 | 40 | 1.26 | 51 | 44.3 vm | 1.4 | - | 24.7 | |
| | 300 | 33 | 1.33 | 44 | 40.5 vm | 1.6 | - | 25.8 | |

vm—given as the volatile matter (%).

This research showed that SS torrefaction at the lowest temperature of 200 °C was sufficient due to the lack of significant return on the *HHV* increase (Figure 10), ash content increase (Figure 8), and energy consumption for the process (Table 2). On the other hand, the torrefied D showed the opposite trend, as *HHV* increased with process temperature and time (Figure 10), i.e., the CSF produced at 300 °C and 30 min (the curve fitted value from the measured time periods) had the highest *HHV* (20 MJ·kg$^{-1}$). This value was comparable to torrefied sawdust at 200 °C (19.6 MJ·kg$^{-1}$) [39] or the torrefied pruned biomass of the Oxytree at 240 °C (20.4 MJ·kg$^{-1}$) [41].

The energetic properties of SS and D-derived CSF were not as high as those associated with other alternative biowaste material used to torrefaction. For example, torrefied spent coffee grounds had an *HHV* of 21–22 MJ·kg$^{-1}$ with an ash content of 1.4% [44], and the *HHV* for the de-oiled seed from biodiesel production was ~23 MJ·kg$^{-1}$ with ash content of ~9.4% [45].

### 4.2. Thermogravimetric Analysis of Raw Materials and Kinetic Parameters of Torrefaction

The reported TGA analyses of SS and D showed that both materials have similar activation energies of 46.7 and 52.2 kJ·mol$^{-1}$, yet different pre-exponential factors of 0.75 and 1.95 s$^{-1}$, respectively (Table 4). Table 4 summarizes the kinetic parameters of materials for which these parameters were determined by the same method as in the study. The higher *k* values were associated with higher decomposition rates and higher mass losses during torrefaction. This was confirmed by *MY*, as *MY* for D was lower than for SS at the same process temperature (Figure 4). The tested materials were more thermally degradable than elephant dung ($k = 1.16 \times 10^{-6}$~$2.73 \times 10^{-5}$ s$^{-1}$) and spent mushroom compost ($k = 1.70 \times 10^{-5}$~$4.60 \times 10^{-5}$ s$^{-1}$), and they were less degradable from lignocellulose materials such pruned Oxytree biomass ($k = 1.43 \times 10^{-5}$~$7.25 \times 10^{-5}$ s$^{-1}$) (Table 4). This was likely due to the *OM* composition. SS and D have less lignin than woody materials. Lignin is harder to decompose than other biomass constituents such as hemicellulose or cellulose.

**Table 4.** Summary of the torrefaction kinetic parameters for different materials.

| Material | Mass, g | Experimental k (200–300 °C), s$^{-1}$ | E$_a$,J·mol$^{-1}$ | A, s$^{-1}$ | OM, % | Ash, % | Reference |
|---|---|---|---|---|---|---|---|
| Sewage sludge | 2.25 | $4.73 \times 10^{-6}$–$3.83 \times 10^{-5}$ | 46.70 | $7.48 \times 10^{-1}$ | 61.9 | 36.3 | - |
| Digestate | 2.25 | $4.91 \times 10^{-6}$–$4.60 \times 10^{-5}$ | 52.23 | $1.94 \times 10^{0}$ | 86.6 | 12.4 | - |
| Sewage sludge | 2.25 | $4.02 \times 10^{-5}$–$6.71 \times 10^{-5}$ | 12.02 * | $6.97 \times 10^{-4}$ * | 59.7 | 40.3 | [38] |
| Elephant dung | 2.25 | $1.16 \times 10^{-6}$–$2.73 \times 10^{-5}$ | 17.70 * | $9.60 \times 10^{-4}$ * | 48.9 | 50.8 | [34] |
| Spent mushroom compost | 2.25 | $1.70 \times 10^{-5}$–$4.60 \times 10^{-5}$ | 21.92 * | $3.90 \times 10^{-3}$ * | 71.6 | 28.4 | [40] |
| Pruning Oxytree biomass | 3.00 | $1.43 \times 10^{-5}$–$7.25 \times 10^{-5}$ * | 36.44 * | $1.53 \times 10^{-1}$ * | 90.2 | 8.1 | [41] |
| RDF | - | $2.11 \times 10^{-3}$–$1.75 \times 10^{-3}$ * | 3.67 * | $3.50 \times 10^{-5}$ * | 85.8 | 13.3 | [39] |

* recalculated in accordance with Section 2.4. TGA of raw material based on means *k* value available in articles: [34,38–41].

The chemical SS composition differed depending on the origin. Hattori and Mukai [46] tested six SS materials with *OM* ranging from 32.3 to 94.1%, and the hemicelluloses, celluloses, and lignin content ranged from 5.1% to 9.8%, 0.2% to 5%, and 9.9% to 29.1%, respectively [46]. The tested D was mainly made from corn (30%), beet pulp (30%), and organic municipal waste (34%), so each constituent of D was a non-lignin material. For example, corn stover is mainly composed of cellulose (~35%), hemicellulose (~20%), and lignin (~12%) [47], and sugar beet pulp is primarily composed of hemicellulose (~23%), cellulose (~22%) and lignin (~2%) [48]. For comparison, wood is typically composed of ~25% hemicelluloses, 45% cellulose, and 25% lignin [49]. Ash content is almost always very low at <5% [50], i.e., over 95% of the mass is organic, and in result, the total amount of lignin was higher than in SS or D where ash decreased the amount of *OM*.

### 4.3. Differential Scanning Calorimetry Analysis

For both tested materials, a DSC analysis began with the endothermic reaction peaks at 102 (or 155) and 85 °C for SS and D, respectively (Figures 15 and 16). Because SS and D were dried before DSC analysis and reactions start at lower temperatures than drying temperature (105 °C), water evaporation could be excluded as a reason for this phenomenon. On the other hand, biomass samples may have absorbed some moisture from the air before the test. In the study of Bryś et al. [51], endothermic peaks were observed in the temperature range of 80–120 °C for dry and wet woody biomass (beech, willow, alder, and spruce). These peaks were assigned to moisture evaporation. For all wet woody biomass,

a large endothermic peak derived from the water was observed, while in the samples after drying, the peak was very small [51].

Chemical composition was not tested during this study; thus, the origin of particular transformations remains unclear. Peaks at 102, 155, and 85 °C were unlikely to belong to the degradation of proteins, fats, or sugars. Protein peaks took place at ~60–100 °C [52]. Fat melting and crystallization peaks were found at lower temperatures than ~40–45 °C [53,54]. In contrast, sugar transitions peaks tend to have sharper shapes than those found in this study and take place at different temperatures, e.g., fructose, glucose, and sucrose melt at 135~156, 159~180, and 194~203 °C, respectively [55].

In general, the charring process is exothermal, whereas volatilization is endothermic [56]. In our study, the results of the DSC analysis did not have an apparent link to the results of the TGA analysis. The occurrence of endothermic reactions did not make any apparent mass changes in the DTG plot (Figure 14). The endothermic reactions ended at ~170 °C, whereas a mass loss in the DTG plot started >200 °C. This might have been a result of insufficient precision in the use of the laboratory balance. The tested D had one exothermic transformation at 327 °C (Figure 16). This transformation may have been a result of lignin charring. A study by Yang et al. [57] revealed that separate DSC analyses of hemicellulose and lignin showed exothermic peaks at 275 and 365 °C, respectively, whereas the thermal degradation of cellulose was endothermic.

The calculated energy needed to heat up a 1 g of the dry mass of the tested materials from 20 to 300 °C in inert conditions was 643 and 492 $J\cdot g^{-1}$ for SS and D, respectively. On the other hand, the energy needed to produce 1 g of CSF in the same conditions (60 min) was, respectively, 882 and 1130 $J\cdot g^{-1}$. This was likely a result of mass losses during torrefaction, i.e., more than 1 g of raw material must be processed to produce 1 g of CSF.

Table 2 shows that the energy balance of dry SS and D torrefaction was energetically self-efficient due to the energy contained in torrgas. The heat of torrgas and $HHV_{torrgas}$ was greater than the energy needed to heat SS and D to the setpoint temperature. Consequently, torrgas can be used as a source of energy for a torrefaction process. Of course, these are only theoretical calculations based on small samples of SS and D that were torrefied in ideal conditions. Scaling up with more complex calculations—that should cover, e.g., heat losses during CSF cooling, the air temperature used to torrgas combustion, equipment efficiency—are still needed. Water evaporation (~2 257 $J\cdot g^{-1}$ at 100 °C and 1 atm) [58] from raw waste should be also included.

In terms of CSF energy content, the calculations showed that the best variant for SS torrefaction was 200 °C and 60 min, where the produced CSF had 14 692 $J\cdot g^{-1}$, whereas, for D, it was 300 °C and 40 min, which produced CSF with 20 082 $J\cdot g^{-1}$. The production of these CSFs consumed 15,368 and 45,918 $J\cdot g^{-1}$ of energy contained in raw SS and D, respectively. The differences between output and input energy increased by energy added to heat a raw material comprised energy that was converted to torrgas. For the best variants of CSF production, the values of energy contained in torrgas (heat and $HHV_{torrgas}$) were 1150 and 27 085 $J\cdot g^{-1}$ for SS and D, respectively.

The best variants of SS and D CSF had *HHV* values of 14.8 and 20 $MJ\cdot kg^{-1}$, respectively. Based on these values, SS and D can be classified in accordance with EN 15359:2012 standard to third and fourth classes, for which *LHV* has to be ≥15 and ≥10 $MJ\cdot kg^{-1}$, respectively. Thus, it is possible that D-derived CSF would be classified as second class (≥20 $MJ\cdot kg^{-1}$); nevertheless, moisture absorbed from the atmosphere during CSF storage can make torrefaction difficult. The content of chlorine and mercury was not measured in this study.

## 5. Conclusions

The following conclusions arise from this research:

- The torrefaction of dry sewage sludge and digestate is energetically self-sufficient.
- Torrefaction improved the higher heating value of the digestate, but it did not improve the *HHV* of sewage sludge. The torrefied digestate had the highest *HHV* = 20 $MJ\cdot kg^{-1}$ under 300 °C and 30 min (the curve fitted from the measured time periods) compared to *HHV* = 18 $MJ\cdot kg^{-1}$ for the

unprocessed digestate. The torrefied sewage sludge had the highest $HHV$ = 14.8 MJ·kg$^{-1}$ under 200 °C and 20 min, as compared to $HHV$ 14.6 MJ·kg$^{-1}$ for raw sewage sludge.

- An unwanted result of torrefaction is an increase in ash content in CSF. A higher ash content results in higher waste production during combustion on the incineration plant. Ash content in the torrefied digestate with the highest $HHV$ was 22%, whereas sewage sludge was 40% ash.
- The kinetics parameters showed that both materials had similar thermal degradability.
- To heat a dried sewage sludge and digestate from 20 to 300 °C, 643 and 492 J·g$^{-1}$ are needed, respectively.
- Approximately 15.4 and 45.9 MJ·kg$^{-1}$ of energy contained in the dry sewage sludge and digestate are needed to produce CSF with the greatest $HHV$, respectively.

This research shows that there is a potential in using D as a substrate for torrefaction and its valorization as an improved fuel source, whereas the potential in using SS for fuel and is questionable due to a lack of $HHV$ increase. The energetic potential of CSF can be enhanced by increasing the density of the material (pelletization), but this process requires additional energy [59]. Due to CSF's low energy value, it seems that it would be more profitable to find another application for this material, e.g., agriculture, because SS-derived CSF has a reduced heavy metal mobility for the reclamation of contaminated sites [60] or as a soil fertilizer [61].

The next step should be to identify the technological parameters for the torrefaction of D on a technical scale and to check the possibilities of further energy densification (e.g., by pelletization). This is important for the investment analysis and technology design of the process on the industrial scale.

**Supplementary Materials:** The following are available online at http://www.mdpi.com/1996-1073/13/12/3161/s1. Excel: data on the fuel properties of carbonized solid fuels (CSF) produced from sewage sludge and digestate.

**Author Contributions:** Conceptualization, K.Ś.; methodology, K.Ś., and P.S.; software, K.Ś.; validation, K.Ś., P.S., and M.H.; formal analysis, K.Ś., M.H., and S.K.; investigation, M.H., P.S., S.K., and K.Ś.; resources, P.S.; data curation, K.Ś.; writing—original draft preparation, K.Ś.; writing—review and editing, K.Ś., S.S.-D., A.B., and J.K.; visualization, K.Ś.; supervision, A.B., and J.A.K. All authors have read and agreed to the published version of the manuscript.

**Funding:** This research was funded by the Polish Ministry of Science and Higher Education (2015–2019), the Diamond Grant Program # 0077/DIA/2015/14. This research was partially supported by the Iowa Agriculture and Home Economics Experiment Station, Ames, Iowa. Project no. IOW05556 (Future Challenges in Animal Production Systems: Seeking Solutions through Focused Facilitation) sponsored by Hatch Act and State of Iowa funds.

**Acknowledgments:** The presented article results were obtained as part of the activity of the leading research team—Waste and Biomass Valorization Group (WBVG), https://www.upwr.edu.pl/research/50121/waste_and_biomass_valorization_group_wbvg.html.

**Conflicts of Interest:** The authors declare no conflict of interest. The funders had no role in the design of the study; in the collection, analyses, or interpretation of data; in the writing of the manuscript, or in the decision to publish the results.

## Appendix A

Appendix A contains a statistical evaluation of empirical models presented in the article. Tables A1–A7 present the evaluations of the intercept and coefficients values presented for particular models. In these tables, standardized B coefficients are presented.

Tables A8–A15 show statistical evaluations of statistically significant differences for particular temperatures and residence times for particular observations.

**Table A1.** Statistical evaluation of model coefficients for the *MY* of CSF from sewage sludge and digestate.

| Material | Intercept/ Coefficient | Value of Intercept/ Coefficient | Standard Error | *p* | Lower Limit of Confidence | Upper Limit of Confidence | Standardized β Coefficient |
|---|---|---|---|---|---|---|---|
| Sewage sludge | $a_1$ | $2.50 \times 10^{-1}$ | $6.67 \times 10^{-1}$ | 0.00 | $-1.22 \times 10^{0}$ | $1.72 \times 10^{0}$ | – |
| | $a_2$ | $5.64 \times 10^{-3}$ | $4.40 \times 10^{-3}$ | 0.00 | $-4.05 \times 10^{-3}$ | $1.53 \times 10^{-2}$ | 2.76 |
| | $a_3$ | $-1.08 \times 10^{-5}$ | $0.00 \times 10^{0}$ | 0.00 | $-1.08 \times 10^{-5}$ | $-1.08 \times 10^{-5}$ | $-2.65$ |
| | $a_4$ | $2.08 \times 10^{-2}$ | $1.89 \times 10^{-2}$ | 0.00 | $-2.08 \times 10^{-2}$ | $6.25 \times 10^{-2}$ | 4.88 |
| | $a_5$ | $-1.12 \times 10^{-4}$ | $1.23 \times 10^{-4}$ | 0.00 | $-3.82 \times 10^{-4}$ | $1.57 \times 10^{-4}$ | $-2.12$ |
| | $a_6$ | $-1.01 \times 10^{-4}$ | $7.48 \times 10^{-5}$ | 0.00 | $-2.65 \times 10^{-4}$ | $6.40 \times 10^{-5}$ | $-6.27$ |
| | $a_7$ | $2.26 \times 10^{-9}$ | $0.00 \times 10^{0}$ | 0.00 | $2.26 \times 10^{-9}$ | $2.26 \times 10^{-9}$ | 3.00 |
| Digestate | $a_1$ | $-1.28 \times 10^{0}$ | $2.48 \times 10^{0}$ | 0.00 | $-6.74 \times 10^{0}$ | $4.17 \times 10^{0}$ | - |
| | $a_2$ | $2.02 \times 10^{-2}$ | $1.64 \times 10^{-2}$ | 0.00 | $-1.58 \times 10^{-2}$ | $5.63 \times 10^{-2}$ | 3.55 |
| | $a_3$ | $-4.31 \times 10^{-5}$ | $2.72 \times 10^{-5}$ | 0.00 | $-1.03 \times 10^{-4}$ | $1.67 \times 10^{-5}$ | $-3.79$ |
| | $a_4$ | $3.57 \times 10^{-2}$ | $7.03 \times 10^{-2}$ | 0.00 | $-1.19 \times 10^{-1}$ | $1.91 \times 10^{-1}$ | 3.00 |
| | $a_5$ | $-1.63 \times 10^{-4}$ | $4.56 \times 10^{-4}$ | 0.00 | $-1.17 \times 10^{-3}$ | $8.40 \times 10^{-4}$ | $-1.10$ |
| | $a_6$ | $-1.94 \times 10^{-4}$ | $2.78 \times 10^{-4}$ | 0.00 | $-8.07 \times 10^{-4}$ | $4.18 \times 10^{-4}$ | $-4.34$ |
| | $a_7$ | $4.54 \times 10^{-9}$ | $0.00 \times 10^{0}$ | 0.00 | $4.54 \times 10^{-9}$ | $4.54 \times 10^{-9}$ | 2.16 |

$MY = a_1 + a_2 \cdot T + a_3 \cdot T^2 + a_4 \cdot t + a_5 \cdot t^2 + a_6 \cdot T \cdot t + a_7 \cdot T^2 \cdot t^2$, *T* ranged from 200 °C to 300 °C, *t* ranged from 20 min to 60 min; more information in the 'CSF Production Method and Process Analysis' section.

**Table A2.** Statistical evaluation of model coefficients for the *EDr* of CSF from sewage sludge and digestate.

| Material | Intercept/ Coefficient | Value of Intercept/ Coefficient | Standard Error | *p* | Lower Limit of Confidence | Upper Limit of Confidence | Standardized β coefficient |
|---|---|---|---|---|---|---|---|
| Sewage sludge | $a_1$ | $2.37 \times 10^{-1}$ | $7.31 \times 10^{-1}$ | 0.00 | $-1.37 \times 10^{0}$ | $1.85 \times 10^{0}$ | - |
| | $a_2$ | $7.06 \times 10^{-3}$ | $4.83 \times 10^{-3}$ | 0.00 | $-3.56 \times 10^{-3}$ | $1.77 \times 10^{-2}$ | 4.04 |
| | $a_3$ | $-1.47 \times 10^{-5}$ | $0.00 \times 10^{0}$ | 0.00 | $-1.47 \times 10^{-5}$ | $-1.47 \times 10^{-5}$ | $-4.20$ |
| | $a_4$ | $3.68 \times 10^{-3}$ | $2.07 \times 10^{-2}$ | 0.00 | $-4.20 \times 10^{-2}$ | $4.93 \times 10^{-2}$ | 1.01 |
| | $a_5$ | $3.02 \times 10^{-5}$ | $1.34 \times 10^{-4}$ | 0.00 | $-2.66 \times 10^{-4}$ | $3.26 \times 10^{-4}$ | 0.67 |
| | $a_6$ | $-3.68 \times 10^{-5}$ | $8.20 \times 10^{-5}$ | 0.00 | $-2.17 \times 10^{-4}$ | $1.44 \times 10^{-4}$ | $-2.67$ |
| | $a_7$ | $3.84 \times 10^{-10}$ | $0.00 \times 10^{0}$ | 0.00 | $3.84 \times 10^{-10}$ | $3.84 \times 10^{-10}$ | 0.59 |
| Digestate | $a_1$ | $1.58 \times 10^{0}$ | $6.37 \times 10^{-1}$ | 0.00 | $1.79 \times 10^{-1}$ | $2.98 \times 10^{0}$ | – |
| | $a_2$ | $-4.26 \times 10^{-3}$ | $4.21 \times 10^{-3}$ | 0.00 | $-1.35 \times 10^{-2}$ | $5.01 \times 10^{-3}$ | $-4.11$ |
| | $a_3$ | $7.42 \times 10^{-6}$ | $0.00 \times 10^{0}$ | 0.00 | $7.42 \times 10^{-6}$ | $7.42 \times 10^{-6}$ | 3.59 |
| | $a_4$ | $-2.29 \times 10^{-2}$ | $1.81 \times 10^{-2}$ | 0.00 | $-6.27 \times 10^{-2}$ | $1.69 \times 10^{-2}$ | $-10.57$ |
| | $a_5$ | $1.67 \times 10^{-4}$ | $1.17 \times 10^{-4}$ | 0.00 | $-9.07 \times 10^{-5}$ | $4.25 \times 10^{-4}$ | 6.24 |
| | $a_6$ | $1.05 \times 10^{-4}$ | $7.16 \times 10^{-5}$ | 0.00 | $-5.22 \times 10^{-5}$ | $2.63 \times 10^{-4}$ | 12.91 |
| | $a_7$ | $-3.27 \times 10^{-9}$ | $0.00 \times 10^{0}$ | 0.00 | $-3.27 \times 10^{-9}$ | $-3.27 \times 10^{-9}$ | $-8.55$ |

$EDr = a_1 + a_2 \cdot T + a_3 \cdot T^2 + a_4 \cdot t + a_5 \cdot t^2 + a_6 \cdot T \cdot t + a_7 \cdot T^2 \cdot t^2$, *T* ranged from 200 °C to 300 °C, *t* ranged from 20 min to 60 min; more information in the 'CSF Production Method and Process Analysis' section.

**Table A3.** Statistical evaluation of model coefficients for the *EY* of CSF from sewage sludge and digestate.

| Material | Intercept/ Coefficient | Value of Intercept/ Coefficient | Standard Error | *p* | Lower Limit of Confidence | Upper Limit of Confidence | Standardized β coefficient |
|---|---|---|---|---|---|---|---|
| Sewage sludge | $a_1$ | $-1.94 \times 10^{-1}$ | $1.09 \times 10^{0}$ | 0.00 | $-2.59 \times 10^{0}$ | $2.20 \times 10^{0}$ | - |
| | $a_2$ | $1.02 \times 10^{-2}$ | $7.19 \times 10^{-3}$ | 0.00 | $-5.62 \times 10^{-3}$ | $2.60 \times 10^{-2}$ | 3.10 |
| | $a_3$ | $-2.08 \times 10^{-5}$ | $1.19 \times 10^{-5}$ | 0.00 | $-4.71 \times 10^{-5}$ | $5.43 \times 10^{-6}$ | $-3.18$ |
| | $a_4$ | $2.06 \times 10^{-2}$ | $3.09 \times 10^{-2}$ | 0.00 | $-4.74 \times 10^{-2}$ | $8.85 \times 10^{-2}$ | 2.99 |
| | $a_5$ | $-7.24 \times 10^{-5}$ | $2.00 \times 10^{-4}$ | 0.00 | $-5.13 \times 10^{-4}$ | $3.68 \times 10^{-4}$ | $-0.85$ |
| | $a_6$ | $-1.18 \times 10^{-4}$ | $1.22 \times 10^{-4}$ | 0.00 | $-3.86 \times 10^{-4}$ | $1.51 \times 10^{-4}$ | $-4.55$ |
| | $a_7$ | $2.35 \times 10^{-9}$ | $0.00 \times 10^{0}$ | 0.00 | $2.35 \times 10^{-9}$ | $2.35 \times 10^{-9}$ | 1.94 |
| Digestate | $a_1$ | $-1.08 \times 10^{0}$ | $2.37 \times 10^{0}$ | 0.00 | $-6.30 \times 10^{0}$ | $4.14 \times 10^{0}$ | – |
| | $a_2$ | $1.90 \times 10^{-2}$ | $1.57 \times 10^{-2}$ | 0.00 | $-1.55 \times 10^{-2}$ | $5.35 \times 10^{-2}$ | 3.48 |
| | $a_3$ | $-4.13 \times 10^{-5}$ | $2.60 \times 10^{-5}$ | 0.00 | $-9.85 \times 10^{-5}$ | $1.60 \times 10^{-5}$ | $-3.79$ |
| | $a_4$ | $2.28 \times 10^{-2}$ | $6.73 \times 10^{-2}$ | 0.00 | $-1.25 \times 10^{-1}$ | $1.71 \times 10^{-1}$ | 1.99 |
| | $a_5$ | $-5.70 \times 10^{-5}$ | $4.36 \times 10^{-4}$ | 0.00 | $-1.02 \times 10^{-3}$ | $9.03 \times 10^{-4}$ | $-0.40$ |
| | $a_6$ | $-1.37 \times 10^{-4}$ | $2.66 \times 10^{-4}$ | 0.00 | $-7.23 \times 10^{-4}$ | $4.49 \times 10^{-4}$ | $-3.19$ |
| | $a_7$ | $2.66 \times 10^{-9}$ | $0.00 \times 10^{0}$ | 0.00 | $2.66 \times 10^{-9}$ | $2.66 \times 10^{-9}$ | 1.32 |

$EY = a_1 + a_2 \cdot T + a_3 \cdot T^2 + a_4 \cdot t + a_5 \cdot t^2 + a_6 \cdot T \cdot t + a_7 \cdot T^2 \cdot t^2$, *T* ranged from 200 °C to 300 °C, *t* ranged from 20 min to 60 min; more information in the 'CSF Production Method and Process Analysis' section.

**Table A4.** Statistical evaluation of model coefficients for the *OM* content in CSF from sewage sludge and digestate.

| Material. | Intercept/ Coefficient | Value of Intercept/ Coefficient | Standard Error | $p$ | Lower Limit of Confidence | Upper Limit of Confidence | Standardized β Coefficient |
|---|---|---|---|---|---|---|---|
| Sewage sludge | $a_1$ | $-2.18 \times 10^{-2}$ | $1.99 \times 10^{-1}$ | 0.00 | $-4.23 \times 10^{-1}$ | $3.79 \times 10^{-1}$ | – |
| | $a_2$ | $5.07 \times 10^{-3}$ | $1.32 \times 10^{-3}$ | 0.00 | $2.42 \times 10^{-3}$ | $7.72 \times 10^{-3}$ | 4.45 |
| | $a_3$ | $-1.01 \times 10^{-5}$ | $0.00 \times 10^{0}$ | 0.00 | $-1.01 \times 10^{-5}$ | $-1.01 \times 10^{-5}$ | $-4.43$ |
| | $a_4$ | $9.46 \times 10^{-3}$ | $5.65 \times 10^{-3}$ | 0.00 | $-1.92 \times 10^{-3}$ | $2.08 \times 10^{-2}$ | 3.97 |
| | $a_5$ | $-4.42 \times 10^{-5}$ | $3.66 \times 10^{-5}$ | 0.00 | $-1.18 \times 10^{-4}$ | $2.95 \times 10^{-5}$ | $-1.50$ |
| | $a_6$ | $-4.62 \times 10^{-5}$ | $2.24 \times 10^{-5}$ | 0.00 | $-9.12 \times 10^{-5}$ | $-1.21 \times 10^{-6}$ | $-5.15$ |
| | $a_7$ | $9.40 \times 10^{-10}$ | $0.00 \times 10^{0}$ | 0.00 | $9.40 \times 10^{-10}$ | $9.40 \times 10^{-10}$ | 2.24 |
| Digestate | $a_1$ | $5.49 \times 10^{-1}$ | $3.08 \times 10^{-1}$ | 0.00 | $-7.04 \times 10^{-2}$ | $1.17 \times 10^{0}$ | – |
| | $a_2$ | $3.37 \times 10^{-3}$ | $2.03 \times 10^{-3}$ | 0.00 | $-7.17 \times 10^{-4}$ | $7.46 \times 10^{-3}$ | 2.33 |
| | $a_3$ | $-8.17 \times 10^{-6}$ | $0.00 \times 10^{0}$ | 0.00 | $-8.17 \times 10^{-6}$ | $-8.17 \times 10^{-6}$ | $-2.83$ |
| | $a_4$ | $1.93 \times 10^{-3}$ | $8.73 \times 10^{-3}$ | 0.00 | $-1.56 \times 10^{-2}$ | $1.95 \times 10^{-2}$ | 0.64 |
| | $a_5$ | $1.06 \times 10^{-5}$ | $5.66 \times 10^{-5}$ | 0.00 | $-1.03 \times 10^{-4}$ | $1.24 \times 10^{-4}$ | 0.28 |
| | $a_6$ | $-1.57 \times 10^{-5}$ | $3.46 \times 10^{-5}$ | 0.00 | $-8.52 \times 10^{-5}$ | $5.38 \times 10^{-5}$ | $-1.38$ |
| | $a_7$ | $1.02 \times 10^{-10}$ | $0.00 \times 10^{0}$ | 0.00 | $1.02 \times 10^{-10}$ | $1.02 \times 10^{-10}$ | 0.19 |

$OM = a_1 + a_2{\cdot}T + a_3{\cdot}T^2 + a_4{\cdot}t + a_5{\cdot}t^2 + a_6{\cdot}T{\cdot}t + a_7{\cdot}T^2{\cdot}t^2$, $T$ ranged from 200 °C to 300 °C, $t$ ranged from 20 min to 60 min; more information in the 'CSF production method and process analysis' section.

**Table A5.** Statistical evaluation of model coefficients for the ash content in CSF from sewage sludge and digestate.

| Material | Intercept/ Coefficient | Value of Intercept/ Coefficient | Standard Error | $p$ | Lower Limit of Confidence | Upper Limit of Confidence | Standardized β Coefficient |
|---|---|---|---|---|---|---|---|
| Sewage sludge | $a_1$ | $9.26 \times 10^{-1}$ | $1.94 \times 10^{-1}$ | 0.00 | $5.36 \times 10^{-1}$ | $1.32 \times 10^{0}$ | – |
| | $a_2$ | $-4.46 \times 10^{-3}$ | $1.28 \times 10^{-3}$ | 0.00 | $-7.03 \times 10^{-3}$ | $-1.88 \times 10^{-3}$ | $-4.15$ |
| | $a_3$ | $8.83 \times 10^{-6}$ | $0.00 \times 10^{0}$ | 0.00 | $8.83 \times 10^{-6}$ | $8.83 \times 10^{-6}$ | 4.12 |
| | $a_4$ | $-8.71 \times 10^{-3}$ | $5.49 \times 10^{-3}$ | 0.00 | $-1.98 \times 10^{-2}$ | $2.34 \times 10^{-3}$ | $-3.88$ |
| | $a_5$ | $3.81 \times 10^{-5}$ | $3.56 \times 10^{-5}$ | 0.00 | $-3.35 \times 10^{-5}$ | $1.10 \times 10^{-4}$ | 1.37 |
| | $a_6$ | $4.31 \times 10^{-5}$ | $2.17 \times 10^{-5}$ | 0.00 | $-6.46 \times 10^{-7}$ | $8.68 \times 10^{-5}$ | 5.10 |
| | $a_7$ | $-8.48 \times 10^{-10}$ | $0.00 \times 10^{0}$ | 0.00 | $-8.48 \times 10^{-10}$ | $-8.48 \times 10^{-10}$ | $-2.14$ |
| Digestate | $a_1$ | $-1.73 \times 10^{-2}$ | $3.26 \times 10^{-1}$ | 0.00 | $-6.74 \times 10^{-1}$ | $6.39 \times 10^{-1}$ | – |
| | $a_2$ | $-5.53 \times 10^{-4}$ | $2.15 \times 10^{-3}$ | 0.00 | $-4.89 \times 10^{-3}$ | $3.78 \times 10^{-3}$ | $-0.38$ |
| | $a_3$ | $4.04 \times 10^{-6}$ | $0.00 \times 10^{0}$ | 0.00 | $4.04 \times 10^{-6}$ | $4.04 \times 10^{-6}$ | 1.37 |
| | $a_4$ | $7.62 \times 10^{-3}$ | $9.25 \times 10^{-3}$ | 0.00 | $-1.10 \times 10^{-2}$ | $2.62 \times 10^{-2}$ | 2.47 |
| | $a_5$ | $-6.43 \times 10^{-5}$ | $6.00 \times 10^{-5}$ | 0.00 | $-1.85 \times 10^{-4}$ | $5.63 \times 10^{-5}$ | $-1.69$ |
| | $a_6$ | $-2.01 \times 10^{-5}$ | $3.66 \times 10^{-5}$ | 0.00 | $-9.37 \times 10^{-5}$ | $5.36 \times 10^{-5}$ | $-1.73$ |
| | $a_7$ | $6.43 \times 10^{-10}$ | $0.00 \times 10^{0}$ | 0.00 | $6.43 \times 10^{-10}$ | $6.43 \times 10^{-10}$ | 1.18 |

Ash $= a_1 + a_2{\cdot}T + a_3{\cdot}T^2 + a_4{\cdot}t + a_5{\cdot}t^2 + a_6{\cdot}T{\cdot}t + a_7{\cdot}T^2{\cdot}t^2$, $T$ ranged from 200 °C to 300 °C, $t$ ranged from 20 min to 60 min; more information in the 'CSF Production Method and Process Analysis' section.

**Table A6.** Statistical evaluation of model coefficients for the *CP* in CSF from sewage sludge and digestate.

| Material | Intercept/ Coefficient | Value of Intercept/ Coefficient | Standard Error | $p$ | Lower Limit of Confidence | Upper Limit of Confidence | Standardized β Coefficient |
|---|---|---|---|---|---|---|---|
| Sewage sludge | $a_1$ | $7.40 \times 10^{-2}$ | $1.94 \times 10^{-1}$ | 0.00 | $-3.15 \times 10^{-1}$ | $4.64 \times 10^{-1}$ | – |
| | $a_2$ | $4.46 \times 10^{-3}$ | $1.28 \times 10^{-3}$ | 0.00 | $1.88 \times 10^{-3}$ | $7.03 \times 10^{-3}$ | 4.15 |
| | $a_3$ | $-8.83 \times 10^{-6}$ | $0.00 \times 10^{0}$ | 0.00 | $-8.83 \times 10^{-6}$ | $-8.83 \times 10^{-6}$ | $-4.12$ |
| | $a_4$ | $8.71 \times 10^{-3}$ | $5.49 \times 10^{-3}$ | 0.00 | $-2.34 \times 10^{-3}$ | $1.98 \times 10^{-2}$ | 3.88 |
| | $a_5$ | $-3.81 \times 10^{-5}$ | $3.56 \times 10^{-5}$ | 0.00 | $-1.10 \times 10^{-4}$ | $3.35 \times 10^{-5}$ | $-1.37$ |
| | $a_6$ | $-4.31 \times 10^{-5}$ | $2.17 \times 10^{-5}$ | 0.00 | $-8.68 \times 10^{-5}$ | $6.46 \times 10^{-7}$ | $-5.10$ |
| | $a_7$ | $8.48 \times 10^{-10}$ | $0.00 \times 10^{0}$ | 0.00 | $8.48 \times 10^{-10}$ | $8.48 \times 10^{-10}$ | 2.14 |
| Digestate | $a_1$ | $1.02 \times 10^{0}$ | $3.26 \times 10^{-1}$ | 0.00 | $3.61 \times 10^{-1}$ | $1.67 \times 10^{0}$ | – |
| | $a_2$ | $5.53 \times 10^{-4}$ | $2.15 \times 10^{-3}$ | 0.00 | $-3.78 \times 10^{-3}$ | $4.89 \times 10^{-3}$ | 0.38 |
| | $a_3$ | $-4.04 \times 10^{-6}$ | $0.00 \times 10^{-0}$ | 0.00 | $-4.04 \times 10^{-6}$ | $-4.04 \times 10^{-6}$ | $-1.37$ |
| | $a_4$ | $-7.62 \times 10^{-3}$ | $9.25 \times 10^{-3}$ | 0.00 | $-2.62 \times 10^{-2}$ | $1.10 \times 10^{-2}$ | $-2.47$ |
| | $a_5$ | $6.43 \times 10^{-5}$ | $6.00 \times 10^{-5}$ | 0.00 | $-5.63 \times 10^{-5}$ | $1.85 \times 10^{-4}$ | 1.69 |
| | $a_6$ | $2.01 \times 10^{-5}$ | $3.66 \times 10^{-5}$ | 0.00 | $-5.36 \times 10^{-5}$ | $9.37 \times 10^{-5}$ | 1.73 |
| | $a_7$ | $-6.43 \times 10^{-10}$ | $0.00 \times 10^{0}$ | 0.00 | $-6.43 \times 10^{-10}$ | $-6.43 \times 10^{-10}$ | $-1.18$ |

$CP = a_1 + a_2{\cdot}T + a_3{\cdot}T^2 + a_4{\cdot}t + a_5{\cdot}t^2 + a_6{\cdot}T{\cdot}t + a_7{\cdot}T^2{\cdot}t^2$, $T$ ranged from 200 °C to 300 °C, $t$ ranged from 20 min to 60 min; more information in the 'CSF Production Method and Process Analysis' section.

**Table A7.** Statistical evaluation of model coefficients for the *HHV* of CSF from sewage sludge and digestate.

| Material | Intercept/ Coefficient | Value of Intercept/ Coefficient | Standard Error | $p$ | Lower Limit of Confidence | Upper Limit of Confidence | Standardized β Coefficient |
|---|---|---|---|---|---|---|---|
| Sewage sludge | $a_1$ | $3.46 \times 10^0$ | $5.95 \times 10^0$ | 0.00 | $-8.51 \times 10^0$ | $1.54 \times 10^1$ | – |
| | $a_2$ | $1.03 \times 10^{-1}$ | $3.93 \times 10^{-2}$ | 0.00 | $2.40 \times 10^{-2}$ | $1.82 \times 10^{-1}$ | 3.94 |
| | $a_3$ | $-2.14 \times 10^{-4}$ | $6.52 \times 10^{-5}$ | 0.00 | $-3.45 \times 10^{-4}$ | $-8.26 \times 10^{-5}$ | $-4.09$ |
| | $a_4$ | $5.37 \times 10^{-2}$ | $1.69 \times 10^{-1}$ | 0.00 | $-2.86 \times 10^{-1}$ | $3.93 \times 10^{-1}$ | 0.98 |
| | $a_5$ | $4.40 \times 10^{-4}$ | $1.09 \times 10^{-3}$ | 0.00 | $-1.76 \times 10^{-3}$ | $2.64 \times 10^{-3}$ | 0.65 |
| | $a_6$ | $-5.37 \times 10^{-4}$ | $6.68 \times 10^{-4}$ | 0.00 | $-1.88 \times 10^{-3}$ | $8.06 \times 10^{-4}$ | $-2.61$ |
| | $a_7$ | $5.60 \times 10^{-9}$ | $0.00 \times 10^0$ | 0.00 | $5.60 \times 10^{-9}$ | $5.60 \times 10^{-9}$ | 0.58 |
| Digestate | $a_1$ | $2.86 \times 10^1$ | $6.95 \times 10^0$ | 0.00 | $1.46 \times 10^1$ | $4.26 \times 10^1$ | - |
| | $a_2$ | $-7.70 \times 10^{-2}$ | $4.59 \times 10^{-2}$ | 0.00 | $-1.69 \times 10^{-1}$ | $1.53 \times 10^{-2}$ | $-3.78$ |
| | $a_3$ | $1.34 \times 10^{-4}$ | $7.62 \times 10^{-6}$ | 0.00 | $-1.91 \times 10^{-6}$ | $2.87 \times 10^{-4}$ | 3.30 |
| | $a_4$ | $-4.15 \times 10^{-1}$ | $1.97 \times 10^{-1}$ | 0.00 | $-8.11 \times 10^{-1}$ | $-1.82 \times 10^{-2}$ | $-9.72$ |
| | $a_5$ | $3.02 \times 10^{-3}$ | $1.28 \times 10^{-3}$ | 0.00 | $4.56 \times 10^{-4}$ | $5.59 \times 10^{-3}$ | 5.73 |
| | $a_6$ | $1.90 \times 10^{-3}$ | $7.80 \times 10^{-4}$ | 0.00 | $3.36 \times 10^{-4}$ | $3.47 \times 10^{-3}$ | 11.87 |
| | $a_7$ | $-5.91 \times 10^{-8}$ | $0.00 \times 10^0$ | 0.00 | $-5.91 \times 10^{-8}$ | $-5.91 \times 10^{-8}$ | $-7.86$ |

$HHV = a_1 + a_2 \cdot T + a_3 \cdot T^2 + a_4 \cdot t + a_5 \cdot t^2 + a_6 \cdot T \cdot t + a_7 \cdot T^2 \cdot t^2$, $T$ ranged from 200 °C to 300 °C, $t$ ranged from 20 min to 60 min; more information in the 'CSF Production Method and Process Analysis' section.

**Table A8.** Analysis of variance for organic matter content of sewage sludge.

| SS, Tukey test for *OM*, a bold font signifies statistically significant difference ($p < 0.05$) | | 200 | 200 | 200 | 220 | 220 | 220 | 240 | 240 | 240 | 260 | 260 | 260 | 280 | 280 | 280 | 300 | 300 | 300 |
|---|---|---|---|---|---|---|---|---|---|---|---|---|---|---|---|---|---|---|---|
| | | 20 | 40 | 60 | 20 | 40 | 60 | 20 | 40 | 60 | 20 | 40 | 60 | 20 | 40 | 60 | 20 | 40 | 60 |
| **200** | 20 | | 0.97 | **0.00** | **0.02** | **0.00** | **0.00** | 1.00 | **0.00** | **0.00** | 0.09 | **0.00** | **0.00** | **0.00** | **0.00** | **0.00** | **0.00** | **0.00** | **0.00** |
| 200 | 40 | 0.97 | | **0.00** | **0.00** | **0.00** | **0.00** | 0.57 | **0.00** | **0.00** | **0.00** | **0.00** | **0.00** | **0.00** | **0.00** | **0.00** | **0.00** | **0.00** | **0.00** |
| 200 | 60 | **0.00** | **0.00** | | 0.29 | 1.00 | 0.87 | **0.00** | 0.08 | **0.00** | 0.06 | **0.00** | **0.02** | **0.00** | **0.00** | **0.00** | **0.00** | **0.00** | **0.00** |
| 220 | 20 | **0.02** | **0.00** | 0.29 | | 0.08 | **0.00** | 0.11 | **0.00** | **0.00** | 1.00 | **0.00** | **0.00** | **0.00** | **0.00** | **0.00** | **0.00** | **0.00** | **0.00** |
| 220 | 40 | **0.00** | **0.00** | 1.00 | 0.08 | | 1.00 | **0.00** | 0.28 | **0.00** | **0.01** | **0.00** | 0.08 | **0.00** | **0.00** | **0.00** | **0.00** | **0.00** | **0.00** |
| 220 | 60 | **0.00** | **0.00** | 0.87 | **0.00** | 1.00 | | **0.00** | 0.97 | **0.01** | **0.00** | **0.00** | 0.73 | **0.00** | **0.00** | **0.00** | **0.00** | **0.00** | **0.00** |
| 240 | 20 | 1.00 | 0.57 | **0.00** | 0.11 | **0.00** | **0.00** | | **0.00** | **0.00** | 0.44 | **0.00** | **0.00** | **0.00** | **0.00** | **0.00** | **0.00** | **0.00** | **0.00** |
| 240 | 40 | **0.00** | **0.00** | 0.08 | **0.00** | 0.28 | 0.97 | **0.00** | | 0.46 | **0.00** | 0.17 | 1.00 | **0.00** | **0.00** | **0.00** | **0.00** | **0.00** | **0.00** |
| 240 | 60 | **0.00** | **0.00** | **0.00** | **0.00** | **0.00** | **0.01** | **0.00** | 0.46 | | **0.00** | 1.00 | 0.84 | **0.03** | **0.00** | **0.00** | **0.00** | **0.00** | **0.00** |
| 260 | 20 | 0.09 | **0.00** | 0.06 | 1.00 | **0.01** | **0.00** | 0.44 | **0.00** | **0.00** | | **0.00** | **0.00** | **0.00** | **0.00** | **0.00** | **0.00** | **0.00** | **0.00** |
| 260 | 40 | **0.00** | **0.00** | **0.00** | **0.00** | **0.00** | **0.00** | **0.00** | 0.17 | 1.00 | **0.00** | | 0.47 | 0.13 | **0.00** | **0.00** | **0.00** | **0.00** | **0.00** |
| 260 | 60 | **0.00** | **0.00** | **0.02** | **0.00** | 0.08 | 0.73 | **0.00** | 1.00 | 0.84 | **0.00** | 0.47 | | **0.00** | **0.00** | **0.00** | **0.00** | **0.00** | **0.00** |
| 280 | 20 | **0.00** | **0.00** | **0.00** | **0.00** | **0.00** | **0.00** | **0.00** | **0.00** | **0.03** | **0.00** | 0.13 | **0.00** | | **0.00** | **0.00** | **0.00** | **0.00** | **0.00** |
| 280 | 40 | **0.00** | **0.00** | **0.00** | **0.00** | **0.00** | **0.00** | **0.00** | **0.00** | **0.00** | **0.00** | **0.00** | **0.00** | **0.00** | | **0.00** | **0.00** | 0.50 | **0.00** |
| 280 | 60 | **0.00** | **0.00** | **0.00** | **0.00** | **0.00** | **0.00** | **0.00** | **0.00** | **0.00** | **0.00** | **0.00** | **0.00** | **0.00** | **0.00** | | **0.00** | **0.00** | 0.39 |
| 300 | 20 | **0.00** | **0.00** | **0.00** | **0.00** | **0.00** | **0.00** | **0.00** | **0.00** | **0.00** | **0.00** | **0.00** | **0.00** | **0.00** | **0.00** | **0.00** | | **0.00** | **0.00** |
| 300 | 40 | **0.00** | **0.00** | **0.00** | **0.00** | **0.00** | **0.00** | **0.00** | **0.00** | **0.00** | **0.00** | **0.00** | **0.00** | **0.00** | 0.50 | **0.00** | **0.00** | | 0.17 |
| 300 | 60 | **0.00** | **0.00** | **0.00** | **0.00** | **0.00** | **0.00** | **0.00** | **0.00** | **0.00** | **0.00** | **0.00** | **0.00** | **0.00** | **0.00** | 0.39 | **0.00** | 0.17 | |

**Table A9.** Analysis of variance for organic matter content of digestate.

| D, Tukey test for *OM*, a bold font signifies statistically significant difference ($p < 0.05$) | | 200 | 200 | 200 | 220 | 220 | 220 | 240 | 240 | 240 | 260 | 260 | 260 | 280 | 280 | 280 | 300 | 300 | 300 |
|---|---|---|---|---|---|---|---|---|---|---|---|---|---|---|---|---|---|---|---|
| | | 20 | 40 | 60 | 20 | 40 | 60 | 20 | 40 | 60 | 20 | 40 | 60 | 20 | 40 | 60 | 20 | 40 | 60 |
| **200** | 20 | | 1.00 | 1.00 | 1.00 | 0.80 | **0.00** | 1.00 | **0.01** | **0.00** | **0.00** | **0.00** | **0.00** | **0.00** | **0.00** | **0.00** | **0.00** | **0.00** | **0.00** |
| 200 | 40 | 1.00 | | 1.00 | 1.00 | 0.89 | **0.01** | 1.00 | **0.02** | **0.00** | **0.00** | **0.00** | **0.00** | **0.00** | **0.00** | **0.00** | **0.00** | **0.00** | **0.00** |
| 200 | 60 | 1.00 | 1.00 | | 1.00 | 1.00 | 0.07 | 0.98 | 0.23 | **0.00** | **0.00** | **0.00** | **0.00** | **0.00** | **0.00** | **0.00** | **0.00** | **0.00** | **0.00** |
| 220 | 20 | 1.00 | 1.00 | 1.00 | | 0.89 | **0.01** | 1.00 | **0.02** | **0.00** | **0.00** | **0.00** | **0.00** | **0.00** | **0.00** | **0.00** | **0.00** | **0.00** | **0.00** |
| 220 | 40 | 0.80 | 0.89 | 1.00 | 0.89 | | 0.42 | 0.60 | 0.78 | **0.00** | **0.00** | **0.00** | **0.01** | **0.00** | **0.00** | **0.00** | **0.00** | **0.00** | **0.00** |
| 220 | 60 | **0.00** | **0.01** | 0.07 | **0.01** | 0.42 | | **0.00** | 1.00 | 0.66 | **0.00** | **0.00** | 0.98 | **0.00** | **0.00** | **0.00** | **0.00** | **0.00** | **0.00** |
| 240 | 20 | 1.00 | 1.00 | 0.98 | 1.00 | 0.60 | **0.00** | | **0.01** | **0.00** | **0.00** | **0.00** | **0.00** | **0.00** | **0.00** | **0.00** | **0.00** | **0.00** | **0.00** |
| 240 | 40 | **0.01** | **0.02** | 0.23 | **0.02** | 0.78 | 1.00 | **0.01** | | 0.31 | **0.00** | **0.00** | 0.79 | **0.00** | **0.00** | **0.00** | **0.00** | **0.00** | **0.00** |
| 240 | 60 | **0.00** | **0.00** | **0.00** | **0.00** | **0.00** | 0.66 | **0.00** | 0.31 | | **0.00** | **0.00** | 1.00 | **0.00** | **0.00** | **0.00** | **0.00** | **0.00** | **0.00** |
| 260 | 20 | **0.00** | **0.00** | **0.00** | **0.00** | **0.00** | **0.00** | **0.00** | **0.00** | **0.00** | | **0.00** | **0.00** | **0.02** | **0.00** | **0.00** | **0.00** | **0.00** | **0.00** |
| 260 | 40 | **0.00** | **0.00** | **0.00** | **0.00** | **0.00** | **0.00** | **0.00** | **0.00** | **0.00** | **0.00** | | **0.00** | 0.93 | **0.00** | **0.00** | **0.00** | **0.00** | **0.00** |
| 260 | 60 | **0.00** | **0.00** | **0.00** | **0.00** | **0.01** | 0.98 | **0.00** | 0.79 | 1.00 | **0.00** | **0.00** | | **0.00** | **0.00** | **0.00** | **0.00** | **0.00** | **0.00** |
| 280 | 20 | **0.00** | **0.00** | **0.00** | **0.00** | **0.00** | **0.00** | **0.00** | **0.00** | **0.00** | **0.02** | 0.93 | **0.00** | | **0.00** | **0.00** | **0.00** | **0.00** | **0.00** |
| 280 | 40 | **0.00** | **0.00** | **0.00** | **0.00** | **0.00** | **0.00** | **0.00** | **0.00** | **0.00** | **0.00** | **0.00** | **0.00** | **0.00** | | **0.00** | **0.00** | 0.08 | 0.67 |
| 280 | 60 | **0.00** | **0.00** | **0.00** | **0.00** | **0.00** | **0.00** | **0.00** | **0.00** | **0.00** | **0.00** | **0.00** | **0.00** | **0.00** | **0.00** | | **0.00** | **0.00** | **0.00** |
| 300 | 20 | **0.00** | **0.00** | **0.00** | **0.00** | **0.00** | **0.00** | **0.00** | **0.00** | **0.00** | **0.00** | **0.00** | **0.00** | **0.00** | **0.00** | **0.00** | | **0.03** | **0.00** |
| 300 | 40 | **0.00** | **0.00** | **0.00** | **0.00** | **0.00** | **0.00** | **0.00** | **0.00** | **0.00** | **0.00** | **0.00** | **0.00** | **0.00** | 0.08 | **0.00** | **0.03** | | **0.00** |
| 300 | 60 | **0.00** | **0.00** | **0.00** | **0.00** | **0.00** | **0.00** | **0.00** | **0.00** | **0.00** | **0.00** | **0.00** | **0.00** | **0.00** | 0.67 | **0.00** | **0.00** | **0.00** | |

**Table A10.** Analysis of variance for ash content of sewage sludge.

| SS, Tukey test for Ash, a bold font signifies statistically significant difference ($p < 0.05$) | | 200 | 200 | 200 | 220 | 220 | 220 | 240 | 240 | 240 | 260 | 260 | 260 | 280 | 280 | 280 | 300 | 300 | 300 |
|---|---|---|---|---|---|---|---|---|---|---|---|---|---|---|---|---|---|---|---|
| | | 20 | 40 | 60 | 20 | 40 | 60 | 20 | 40 | 60 | 20 | 40 | 60 | 20 | 40 | 60 | 20 | 40 | 60 |
| **200** | 20 | | 0.98 | **0.00** | 0.16 | **0.00** | **0.00** | 1.00 | **0.00** | **0.00** | 0.09 | **0.00** | **0.00** | **0.00** | **0.00** | **0.00** | **0.00** | **0.00** | **0.00** |
| 200 | 40 | 0.98 | | **0.00** | **0.00** | **0.00** | **0.00** | 0.76 | **0.00** | **0.00** | **0.00** | **0.00** | **0.00** | **0.00** | **0.00** | **0.00** | **0.00** | **0.00** | **0.00** |
| 200 | 60 | **0.00** | **0.00** | | 0.87 | 0.94 | 0.15 | **0.01** | **0.00** | **0.00** | 0.96 | **0.00** | **0.00** | **0.00** | **0.00** | **0.00** | **0.00** | **0.00** | **0.00** |
| 220 | 20 | 0.16 | **0.00** | 0.87 | | 0.05 | **0.00** | 0.48 | **0.00** | **0.00** | 1.00 | **0.00** | **0.00** | **0.00** | **0.00** | **0.00** | **0.00** | **0.00** | **0.00** |
| 220 | 40 | **0.00** | **0.00** | 0.94 | 0.05 | | 0.98 | **0.00** | 0.17 | **0.02** | 0.10 | **0.00** | **0.04** | **0.00** | **0.00** | **0.00** | **0.00** | **0.00** | **0.00** |
| 220 | 60 | **0.00** | **0.00** | 0.15 | **0.00** | 0.98 | | **0.00** | 0.96 | 0.45 | **0.00** | 0.17 | 0.63 | **0.00** | **0.00** | **0.00** | **0.00** | **0.00** | **0.00** |
| 240 | 20 | 1.00 | 0.76 | **0.01** | 0.48 | **0.00** | **0.00** | | **0.00** | **0.00** | 0.31 | **0.00** | **0.00** | **0.00** | **0.00** | **0.00** | **0.00** | **0.00** | **0.00** |
| 240 | 40 | **0.00** | **0.00** | **0.00** | **0.00** | 0.17 | 0.96 | **0.00** | | 1.00 | **0.00** | 0.98 | 1.00 | **0.00** | **0.00** | **0.00** | **0.00** | **0.00** | **0.00** |
| 240 | 60 | **0.00** | **0.00** | **0.00** | **0.00** | **0.02** | 0.45 | **0.00** | 1.00 | | **0.00** | 1.00 | 1.00 | **0.00** | **0.00** | **0.00** | **0.00** | **0.00** | **0.00** |
| 260 | 20 | 0.09 | **0.00** | 0.96 | 1.00 | 0.10 | **0.00** | 0.31 | **0.00** | **0.00** | | **0.00** | **0.00** | **0.00** | **0.00** | **0.00** | **0.00** | **0.00** | **0.00** |
| 260 | 40 | **0.00** | **0.00** | **0.00** | **0.00** | **0.00** | 0.17 | **0.00** | 0.98 | 1.00 | **0.00** | | 1.00 | **0.02** | **0.00** | **0.00** | **0.00** | **0.00** | **0.00** |
| 260 | 60 | **0.00** | **0.00** | **0.00** | **0.00** | **0.04** | 0.63 | **0.00** | 1.00 | 1.00 | **0.00** | 1.00 | | **0.00** | **0.00** | **0.00** | **0.00** | **0.00** | **0.00** |
| 280 | 20 | **0.00** | **0.00** | **0.00** | **0.00** | **0.00** | **0.00** | **0.00** | **0.00** | **0.00** | **0.00** | **0.02** | **0.00** | | **0.00** | **0.00** | **0.00** | **0.00** | **0.00** |
| 280 | 40 | **0.00** | **0.00** | **0.00** | **0.00** | **0.00** | **0.00** | **0.00** | **0.00** | **0.00** | **0.00** | **0.00** | **0.00** | **0.00** | | **0.00** | **0.00** | 1.00 | **0.03** |
| 280 | 60 | **0.00** | **0.00** | **0.00** | **0.00** | **0.00** | **0.00** | **0.00** | **0.00** | **0.00** | **0.00** | **0.00** | **0.00** | **0.00** | **0.00** | | **0.00** | **0.00** | 0.05 |
| 300 | 20 | **0.00** | **0.00** | **0.00** | **0.00** | **0.00** | **0.00** | **0.00** | **0.00** | **0.00** | **0.00** | **0.00** | **0.00** | **0.00** | **0.00** | **0.00** | | **0.00** | **0.00** |
| 300 | 40 | **0.00** | **0.00** | **0.00** | **0.00** | **0.00** | **0.00** | **0.00** | **0.00** | **0.00** | **0.00** | **0.00** | **0.00** | **0.00** | 1.00 | **0.00** | **0.00** | | 0.32 |
| 300 | 60 | **0.00** | **0.00** | **0.00** | **0.00** | **0.00** | **0.00** | **0.00** | **0.00** | **0.00** | **0.00** | **0.00** | **0.00** | **0.00** | **0.03** | 0.05 | **0.00** | 0.32 | |

**Table A11.** Analysis of variance for ash content of digestate.

| D, Tukey test for Ash, a bold font signifies statistically significant difference ($p < 0.05$) | | 200 | 200 | 200 | 220 | 220 | 220 | 240 | 240 | 240 | 260 | 260 | 260 | 280 | 280 | 280 | 300 | 300 | 300 |
|---|---|---|---|---|---|---|---|---|---|---|---|---|---|---|---|---|---|---|---|
| | | 20 | 40 | 60 | 20 | 40 | 60 | 20 | 40 | 60 | 20 | 40 | 60 | 20 | 40 | 60 | 20 | 40 | 60 |
| **200** | 20 | | 0.25 | 0.09 | 0.17 | **0.02** | **0.00** | 0.17 | **0.01** | **0.00** | **0.00** | **0.00** | **0.00** | **0.00** | **0.00** | **0.00** | **0.00** | **0.00** | **0.00** |
| 200 | 40 | 0.25 | | 1.00 | 1.00 | 1.00 | 0.87 | 1.00 | 0.99 | 0.58 | **0.01** | **0.00** | 0.71 | **0.00** | **0.00** | **0.00** | **0.00** | **0.00** | **0.00** |
| 200 | 60 | 0.09 | 1.00 | | 1.00 | 1.00 | 0.99 | 1.00 | 1.00 | 0.86 | **0.02** | **0.00** | 0.94 | **0.00** | **0.00** | **0.00** | **0.00** | **0.00** | **0.00** |
| 220 | 20 | 0.17 | 1.00 | 1.00 | | 1.00 | 0.94 | 1.00 | 1.00 | 0.71 | **0.01** | **0.00** | 0.83 | **0.00** | **0.00** | **0.00** | **0.00** | **0.00** | **0.00** |
| 220 | 40 | **0.02** | 1.00 | 1.00 | 1.00 | | 1.00 | 1.00 | 1.00 | 1.00 | 0.12 | **0.00** | 1.00 | **0.00** | **0.00** | **0.00** | **0.00** | **0.00** | **0.00** |
| 220 | 60 | **0.00** | 0.87 | 0.99 | 0.94 | 1.00 | | 0.93 | 1.00 | 1.00 | 0.50 | **0.00** | 1.00 | **0.02** | **0.00** | **0.00** | **0.00** | **0.00** | **0.00** |
| 240 | 20 | 0.17 | 1.00 | 1.00 | 1.00 | 1.00 | 0.93 | | 1.00 | 0.70 | **0.01** | **0.00** | 0.82 | **0.00** | **0.00** | **0.00** | **0.00** | **0.00** | **0.00** |
| 240 | 40 | **0.01** | 0.99 | 1.00 | 1.00 | 1.00 | 1.00 | 1.00 | | 1.00 | 0.23 | **0.00** | 1.00 | **0.00** | **0.00** | **0.00** | **0.00** | **0.00** | **0.00** |
| 240 | 60 | **0.00** | 0.58 | 0.86 | 0.71 | 1.00 | 1.00 | 0.70 | 1.00 | | 0.81 | **0.02** | 1.00 | 0.06 | **0.00** | **0.00** | **0.00** | **0.00** | **0.00** |
| 260 | 20 | **0.00** | **0.01** | **0.02** | **0.01** | 0.12 | 0.50 | **0.01** | 0.23 | 0.81 | | 0.81 | 0.69 | 0.97 | **0.00** | **0.00** | **0.04** | **0.00** | **0.00** |
| 260 | 40 | **0.00** | **0.00** | **0.00** | **0.00** | **0.00** | **0.00** | **0.00** | **0.00** | **0.02** | 0.81 | | **0.01** | 1.00 | **0.01** | **0.00** | 0.95 | 0.10 | **0.00** |
| 260 | 60 | **0.00** | 0.71 | 0.94 | 0.83 | 1.00 | 1.00 | 0.82 | 1.00 | 1.00 | 0.69 | **0.01** | | **0.04** | **0.00** | **0.00** | 0.74 | **0.03** | **0.00** |
| 280 | 20 | **0.00** | **0.00** | **0.00** | **0.00** | **0.00** | **0.02** | **0.00** | **0.00** | 0.06 | 0.97 | 1.00 | **0.04** | | **0.00** | **0.00** | 0.74 | **0.03** | **0.00** |
| 280 | 40 | **0.00** | **0.00** | **0.00** | **0.00** | **0.00** | **0.00** | **0.00** | **0.00** | **0.00** | **0.00** | **0.01** | **0.00** | **0.00** | | 0.16 | 0.41 | 1.00 | 1.00 |
| 280 | 60 | **0.00** | **0.00** | **0.00** | **0.00** | **0.00** | **0.00** | **0.00** | **0.00** | **0.00** | **0.00** | **0.00** | **0.00** | **0.00** | 0.16 | | **0.00** | **0.01** | 0.66 |
| 300 | 20 | **0.00** | **0.00** | **0.00** | **0.00** | **0.00** | **0.00** | **0.00** | **0.00** | **0.00** | **0.04** | 0.95 | **0.00** | 0.74 | 0.41 | **0.00** | | 0.95 | 0.07 |
| 300 | 40 | **0.00** | **0.00** | **0.00** | **0.00** | **0.00** | **0.00** | **0.00** | **0.00** | **0.00** | **0.00** | 0.10 | **0.00** | **0.03** | 1.00 | **0.01** | 0.95 | | 0.89 |
| 300 | 60 | **0.00** | **0.00** | **0.00** | **0.00** | **0.00** | **0.00** | **0.00** | **0.00** | **0.00** | **0.00** | **0.00** | **0.00** | **0.00** | 1.00 | 0.66 | 0.07 | 0.89 | |

**Table A12.** Analysis of variance for combustible parts content of sewage sludge.

| SS, Tukey test for *CP*, a bold font signifies statistically significant difference (*p* < 0.05) | | 200 | 200 | 200 | 220 | 220 | 220 | 240 | 240 | 240 | 260 | 260 | 260 | 280 | 280 | 280 | 300 | 300 | 300 |
|---|---|---|---|---|---|---|---|---|---|---|---|---|---|---|---|---|---|---|---|
| | | 20 | 40 | 60 | 20 | 40 | 60 | 20 | 40 | 60 | 20 | 40 | 60 | 20 | 40 | 60 | 20 | 40 | 60 |
| **200** | 20 | | 0.98 | **0.00** | 0.16 | **0.00** | **0.00** | 1.00 | **0.00** | **0.00** | 0.09 | **0.00** | **0.00** | **0.00** | **0.00** | **0.00** | **0.00** | **0.00** | **0.00** |
| 200 | 40 | 0.98 | | **0.00** | **0.00** | **0.00** | **0.00** | 0.76 | **0.00** | **0.00** | **0.00** | **0.00** | **0.00** | **0.00** | **0.00** | **0.00** | **0.00** | **0.00** | **0.00** |
| 200 | 60 | **0.00** | **0.00** | | 0.87 | 0.94 | 0.15 | **0.01** | **0.00** | **0.00** | 0.96 | **0.00** | **0.00** | **0.00** | **0.00** | **0.00** | **0.00** | **0.00** | **0.00** |
| 220 | 20 | 0.16 | **0.00** | 0.87 | | 0.05 | **0.00** | 0.48 | **0.00** | **0.00** | 1.00 | **0.00** | **0.00** | **0.00** | **0.00** | **0.00** | **0.00** | **0.00** | **0.00** |
| 220 | 40 | **0.00** | **0.00** | 0.94 | 0.05 | | 0.98 | **0.00** | 0.17 | **0.02** | 0.10 | **0.00** | **0.04** | **0.00** | **0.00** | **0.00** | **0.00** | **0.00** | **0.00** |
| 220 | 60 | **0.00** | **0.00** | 0.15 | **0.00** | 0.98 | | **0.00** | 0.96 | 0.45 | **0.00** | 0.17 | 0.63 | **0.00** | **0.00** | **0.00** | **0.00** | **0.00** | **0.00** |
| 240 | 20 | 1.00 | 0.76 | **0.01** | 0.48 | **0.00** | **0.00** | | **0.00** | **0.00** | 0.31 | **0.00** | **0.00** | **0.00** | **0.00** | **0.00** | **0.00** | **0.00** | **0.00** |
| 240 | 40 | **0.00** | **0.00** | **0.00** | **0.00** | 0.17 | 0.96 | **0.00** | | 1.00 | **0.00** | 0.98 | 1.00 | **0.00** | **0.00** | **0.00** | **0.00** | **0.00** | **0.00** |
| 240 | 60 | **0.00** | **0.00** | **0.00** | **0.00** | **0.02** | 0.45 | **0.00** | 1.00 | | **0.00** | 1.00 | 1.00 | **0.00** | **0.00** | **0.00** | **0.00** | **0.00** | **0.00** |
| 260 | 20 | 0.09 | **0.00** | 0.96 | 1.00 | 0.10 | **0.00** | 0.31 | **0.00** | **0.00** | | **0.00** | **0.00** | **0.00** | **0.00** | **0.00** | **0.00** | **0.00** | **0.00** |
| 260 | 40 | **0.00** | **0.00** | **0.00** | **0.00** | **0.00** | 0.17 | **0.00** | 0.98 | 1.00 | **0.00** | | 1.00 | **0.02** | **0.00** | **0.00** | **0.00** | **0.00** | **0.00** |
| 260 | 60 | **0.00** | **0.00** | **0.00** | **0.00** | **0.04** | 0.63 | **0.00** | 1.00 | 1.00 | **0.00** | 1.00 | | **0.00** | **0.00** | **0.00** | **0.00** | **0.00** | **0.00** |
| 280 | 20 | **0.00** | **0.00** | **0.00** | **0.00** | **0.00** | **0.00** | **0.00** | **0.00** | **0.00** | **0.00** | **0.02** | **0.00** | | **0.00** | **0.00** | **0.00** | **0.00** | **0.00** |
| 280 | 40 | **0.00** | **0.00** | **0.00** | **0.00** | **0.00** | **0.00** | **0.00** | **0.00** | **0.00** | **0.00** | **0.00** | **0.00** | **0.00** | | **0.00** | **0.00** | 1.00 | **0.03** |
| 280 | 60 | **0.00** | **0.00** | **0.00** | **0.00** | **0.00** | **0.00** | **0.00** | **0.00** | **0.00** | **0.00** | **0.00** | **0.00** | **0.00** | **0.00** | | **0.00** | **0.00** | **0.05** |
| 300 | 20 | **0.00** | **0.00** | **0.00** | **0.00** | **0.00** | **0.00** | **0.00** | **0.00** | **0.00** | **0.00** | **0.00** | **0.00** | **0.00** | **0.00** | **0.00** | | **0.00** | **0.00** |
| 300 | 40 | **0.00** | **0.00** | **0.00** | **0.00** | **0.00** | **0.00** | **0.00** | **0.00** | **0.00** | **0.00** | **0.00** | **0.00** | **0.00** | 1.00 | **0.00** | **0.00** | | 0.32 |
| 300 | 60 | **0.00** | **0.00** | **0.00** | **0.00** | **0.00** | **0.00** | **0.00** | **0.00** | **0.00** | **0.00** | **0.00** | **0.00** | **0.00** | **0.03** | **0.05** | **0.00** | 0.32 | |

**Table A13.** Analysis of variance for combustible parts content of digestate.

| D, Tukey test for *CP*, a bold font signifies statistically significant difference (*p* < 0.05) | | 200 | 200 | 200 | 220 | 220 | 220 | 240 | 240 | 240 | 260 | 260 | 260 | 280 | 280 | 280 | 300 | 300 | 300 |
|---|---|---|---|---|---|---|---|---|---|---|---|---|---|---|---|---|---|---|---|
| | | 20 | 40 | 60 | 20 | 40 | 60 | 20 | 40 | 60 | 20 | 40 | 60 | 20 | 40 | 60 | 20 | 40 | 60 |
| **200** | 20 | | 0.25 | 0.09 | 0.17 | **0.02** | **0.00** | 0.17 | **0.01** | **0.00** | **0.00** | **0.00** | **0.00** | **0.00** | **0.00** | **0.00** | **0.00** | **0.00** | **0.00** |
| 200 | 40 | 0.25 | | 1.00 | 1.00 | 1.00 | 0.87 | 1.00 | 0.99 | 0.58 | **0.01** | **0.00** | 0.71 | **0.00** | **0.00** | **0.00** | **0.00** | **0.00** | **0.00** |
| 200 | 60 | 0.09 | 1.00 | | 1.00 | 1.00 | 0.99 | 1.00 | 1.00 | 0.86 | **0.02** | **0.00** | 0.94 | **0.00** | **0.00** | **0.00** | **0.00** | **0.00** | **0.00** |
| 220 | 20 | 0.17 | 1.00 | 1.00 | | 1.00 | 0.94 | 1.00 | 1.00 | 0.71 | **0.01** | **0.00** | 0.83 | **0.00** | **0.00** | **0.00** | **0.00** | **0.00** | **0.00** |
| 220 | 40 | **0.02** | 1.00 | 1.00 | 1.00 | | 1.00 | 1.00 | 1.00 | 1.00 | 0.12 | **0.00** | 1.00 | **0.00** | **0.00** | **0.00** | **0.00** | **0.00** | **0.00** |
| 220 | 60 | **0.00** | 0.87 | 0.99 | 0.94 | 1.00 | | 0.93 | 1.00 | 1.00 | 0.50 | **0.00** | 1.00 | **0.02** | **0.00** | **0.00** | **0.00** | **0.00** | **0.00** |
| 240 | 20 | 0.17 | 1.00 | 1.00 | 1.00 | 1.00 | 0.93 | | 1.00 | 0.70 | **0.01** | **0.00** | 0.82 | **0.00** | **0.00** | **0.00** | **0.00** | **0.00** | **0.00** |
| 240 | 40 | **0.01** | 0.99 | 1.00 | 1.00 | 1.00 | 1.00 | 1.00 | | 1.00 | 0.23 | **0.00** | 1.00 | **0.00** | **0.00** | **0.00** | **0.00** | **0.00** | **0.00** |
| 240 | 60 | **0.00** | 0.58 | 0.86 | 0.71 | 1.00 | 1.00 | 0.70 | 1.00 | | 0.81 | **0.02** | 1.00 | 0.06 | **0.00** | **0.00** | **0.00** | **0.00** | **0.00** |
| 260 | 20 | **0.00** | **0.01** | **0.02** | **0.01** | 0.12 | 0.50 | **0.01** | 0.23 | 0.81 | | 0.81 | 0.69 | 0.97 | **0.00** | **0.00** | **0.04** | **0.00** | **0.00** |
| 260 | 40 | **0.00** | **0.00** | **0.00** | **0.00** | **0.00** | **0.00** | **0.00** | **0.00** | **0.02** | 0.81 | | **0.01** | 1.00 | **0.01** | **0.00** | 0.95 | 0.10 | **0.00** |
| 260 | 60 | **0.00** | 0.71 | 0.94 | 0.83 | 1.00 | 1.00 | 0.82 | 1.00 | 1.00 | 0.69 | **0.01** | | **0.04** | **0.00** | **0.00** | **0.00** | **0.00** | **0.00** |
| 280 | 20 | **0.00** | **0.00** | **0.00** | **0.00** | **0.00** | **0.02** | **0.00** | **0.00** | 0.06 | 0.97 | 1.00 | **0.04** | | **0.00** | **0.00** | 0.74 | **0.03** | **0.00** |
| 280 | 40 | **0.00** | **0.00** | **0.00** | **0.00** | **0.00** | **0.00** | **0.00** | **0.00** | **0.00** | **0.00** | **0.01** | **0.00** | **0.00** | | 0.16 | 0.41 | 1.00 | 1.00 |
| 280 | 60 | **0.00** | **0.00** | **0.00** | **0.00** | **0.00** | **0.00** | **0.00** | **0.00** | **0.00** | **0.00** | **0.00** | **0.00** | **0.00** | 0.16 | | **0.00** | **0.01** | 0.66 |
| 300 | 20 | **0.00** | **0.00** | **0.00** | **0.00** | **0.00** | **0.00** | **0.00** | **0.00** | **0.00** | **0.04** | 0.95 | **0.00** | 0.74 | 0.41 | **0.00** | | 0.95 | 0.07 |
| 300 | 40 | **0.00** | **0.00** | **0.00** | **0.00** | **0.00** | **0.00** | **0.00** | **0.00** | **0.00** | **0.00** | 0.10 | **0.00** | **0.03** | 1.00 | **0.01** | 0.95 | | 0.89 |
| 300 | 60 | **0.00** | **0.00** | **0.00** | **0.00** | **0.00** | **0.00** | **0.00** | **0.00** | **0.00** | **0.00** | **0.00** | **0.00** | **0.00** | 1.00 | 0.66 | 0.07 | 0.89 | |

**Table A14.** Analysis of variance for HHV of sewage sludge.

| SS, Tukey test for *HHV*, a bold font signifies statistically significant difference ($p < 0.05$) | | 200 20 | 200 40 | 200 60 | 220 20 | 220 40 | 220 60 | 240 20 | 240 40 | 240 60 | 260 20 | 260 40 | 260 60 | 280 20 | 280 40 | 280 60 | 300 20 | 300 40 | 300 60 |
|---|---|---|---|---|---|---|---|---|---|---|---|---|---|---|---|---|---|---|---|
| **200** | 20 | | 0.46 | 1.00 | 0.99 | 0.88 | 0.32 | 1.00 | **0.01** | 0.51 | 0.33 | **0.02** | 0.12 | **0.01** | **0.00** | **0.00** | **0.00** | **0.00** | **0.00** |
| 200 | 40 | 0.46 | | 1.00 | 1.00 | 1.00 | 1.00 | 0.94 | 0.90 | 1.00 | 1.00 | 0.98 | 1.00 | 0.93 | **0.00** | **0.00** | **0.01** | **0.00** | **0.00** |
| 200 | 60 | 1.00 | 1.00 | | 1.00 | 1.00 | 0.98 | 1.00 | 0.18 | 1.00 | 0.98 | 0.34 | 0.81 | 0.21 | **0.00** | **0.00** | **0.00** | **0.00** | **0.00** |
| 220 | 20 | 0.99 | 1.00 | 1.00 | | 1.00 | 0.99 | 1.00 | 0.21 | 1.00 | 0.99 | 0.39 | 0.86 | 0.25 | **0.00** | **0.00** | **0.00** | **0.00** | **0.00** |
| 220 | 40 | 0.88 | 1.00 | 1.00 | 1.00 | | 1.00 | 1.00 | 0.50 | 1.00 | 1.00 | 0.74 | 0.99 | 0.57 | **0.00** | **0.00** | **0.00** | **0.00** | **0.00** |
| 220 | 60 | 0.32 | 1.00 | 0.98 | 0.99 | 1.00 | | 0.86 | 0.97 | 1.00 | 1.00 | 1.00 | 1.00 | 0.98 | **0.00** | **0.00** | **0.01** | **0.00** | **0.00** |
| 240 | 20 | 1.00 | 0.94 | 1.00 | 1.00 | 1.00 | 0.86 | | 0.07 | 0.96 | 0.87 | 0.15 | 0.54 | 0.09 | **0.00** | **0.00** | **0.00** | **0.00** | **0.00** |
| 240 | 40 | **0.01** | 0.90 | 0.18 | 0.21 | 0.50 | 0.97 | 0.07 | | 0.88 | 0.96 | 1.00 | 1.00 | 1.00 | **0.00** | **0.00** | 0.39 | **0.00** | **0.00** |
| 240 | 60 | 0.51 | 1.00 | 1.00 | 1.00 | 1.00 | 1.00 | 0.96 | 0.88 | | 1.00 | 0.98 | 1.00 | 0.91 | **0.00** | **0.00** | **0.00** | **0.00** | **0.00** |
| 260 | 20 | 0.33 | 1.00 | 0.98 | 0.99 | 1.00 | 1.00 | 0.87 | 0.96 | 1.00 | | 1.00 | 1.00 | 0.98 | **0.00** | **0.00** | **0.01** | **0.00** | **0.00** |
| 260 | 40 | **0.02** | 0.98 | 0.34 | 0.39 | 0.74 | 1.00 | 0.15 | 1.00 | 0.98 | 1.00 | | 1.00 | 1.00 | **0.00** | **0.00** | 0.20 | **0.00** | **0.00** |
| 260 | 60 | 0.12 | 1.00 | 0.81 | 0.86 | 0.99 | 1.00 | 0.54 | 1.00 | 1.00 | 1.00 | 1.00 | | 1.00 | **0.00** | **0.00** | **0.04** | **0.00** | **0.00** |
| 280 | 20 | **0.01** | 0.93 | 0.21 | 0.25 | 0.57 | 0.98 | 0.09 | 1.00 | 0.91 | 0.98 | 1.00 | 1.00 | | **0.00** | **0.00** | 0.33 | **0.00** | **0.00** |
| 280 | 40 | **0.00** | **0.00** | **0.00** | **0.00** | **0.00** | **0.00** | **0.00** | **0.00** | **0.00** | **0.00** | **0.00** | **0.00** | **0.00** | | **0.02** | **0.04** | 0.97 | 1.00 |
| 280 | 60 | **0.00** | **0.00** | **0.00** | **0.00** | **0.00** | **0.00** | **0.00** | **0.00** | **0.00** | **0.00** | **0.00** | **0.00** | **0.00** | **0.02** | | **0.00** | 0.51 | 0.24 |
| 300 | 20 | **0.00** | **0.01** | **0.00** | **0.00** | **0.00** | **0.01** | **0.00** | 0.39 | **0.00** | **0.01** | 0.20 | **0.04** | 0.33 | **0.04** | **0.00** | | **0.00** | **0.00** |
| 300 | 40 | **0.00** | **0.00** | **0.00** | **0.00** | **0.00** | **0.00** | **0.00** | **0.00** | **0.00** | **0.00** | **0.00** | **0.00** | **0.00** | 0.97 | 0.51 | **0.00** | | 1.00 |
| 300 | 60 | **0.00** | **0.00** | **0.00** | **0.00** | **0.00** | **0.00** | **0.00** | **0.00** | **0.00** | **0.00** | **0.00** | **0.00** | **0.00** | 1.00 | 0.24 | **0.00** | 1.00 | |

**Table A15.** Analysis of variance for HHV of digestate.

| D, Tukey test for *HHV*, a bold font signifies statistically significant difference ($p < 0.05$) | | 200 20 | 200 40 | 200 60 | 220 20 | 220 40 | 220 60 | 240 20 | 240 40 | 240 60 | 260 20 | 260 40 | 260 60 | 280 20 | 280 40 | 280 60 | 300 20 | 300 40 | 300 60 |
|---|---|---|---|---|---|---|---|---|---|---|---|---|---|---|---|---|---|---|---|
| **200** | 20 | | 1.00 | 0.87 | 0.99 | 0.65 | 0.14 | 1.00 | 1.00 | 0.17 | 1.00 | **0.00** | 0.07 | **0.00** | **0.00** | 1.00 | **0.00** | **0.00** | 0.56 |
| 200 | 40 | 1.00 | | 0.83 | 0.97 | 0.59 | 0.11 | 1.00 | 1.00 | 0.14 | 1.00 | **0.00** | 0.06 | **0.00** | **0.00** | 1.00 | **0.00** | **0.00** | 0.49 |
| 200 | 60 | 0.87 | 0.83 | | 1.00 | 1.00 | 0.99 | 0.47 | 1.00 | 1.00 | 0.89 | 0.05 | 0.96 | **0.00** | 0.07 | 1.00 | 0.22 | **0.01** | 1.00 |
| 220 | 20 | 0.99 | 0.97 | 1.00 | | 1.00 | 0.93 | 0.77 | 1.00 | 0.95 | 0.99 | **0.02** | 0.79 | **0.00** | **0.02** | 1.00 | 0.09 | **0.00** | 1.00 |
| 220 | 40 | 0.65 | 0.59 | 1.00 | 1.00 | | 1.00 | 0.25 | 1.00 | 1.00 | 0.67 | 0.13 | 1.00 | **0.00** | 0.17 | 1.00 | 0.43 | **0.03** | 1.00 |
| 220 | 60 | 0.14 | 0.11 | 0.99 | 0.93 | 1.00 | | **0.03** | 0.69 | 1.00 | 0.15 | 0.63 | 1.00 | **0.03** | 0.72 | 0.69 | 0.95 | 0.25 | 1.00 |
| 240 | 20 | 1.00 | 1.00 | 0.47 | 0.77 | 0.25 | **0.03** | | 0.96 | **0.04** | 1.00 | **0.00** | **0.01** | **0.00** | **0.00** | 0.96 | **0.00** | **0.00** | 0.19 |
| 240 | 40 | 1.00 | 1.00 | 1.00 | 1.00 | 1.00 | 0.69 | 0.96 | | 0.75 | 1.00 | **0.00** | 0.49 | **0.00** | **0.01** | 1.00 | **0.03** | **0.00** | 0.99 |
| 240 | 60 | 0.17 | 0.14 | 1.00 | 0.95 | 1.00 | 1.00 | **0.04** | 0.75 | | 0.18 | 0.57 | 1.00 | **0.03** | 0.65 | 0.75 | 0.92 | 0.21 | 1.00 |
| 260 | 20 | 1.00 | 1.00 | 0.89 | 0.99 | 0.67 | 0.15 | 1.00 | 1.00 | 0.18 | | **0.00** | 0.08 | **0.00** | **0.00** | 1.00 | **0.00** | **0.00** | 0.58 |
| 260 | 40 | **0.00** | **0.00** | 0.05 | **0.02** | 0.13 | 0.63 | **0.00** | **0.00** | 0.57 | **0.00** | | 0.82 | 0.98 | 1.00 | **0.00** | 1.00 | 1.00 | 0.18 |
| 260 | 60 | 0.07 | 0.06 | 0.96 | 0.79 | 1.00 | 1.00 | **0.01** | 0.49 | 1.00 | 0.08 | 0.82 | | 0.07 | 0.88 | 0.49 | 0.99 | 0.42 | 1.00 |
| 280 | 20 | **0.00** | **0.00** | **0.00** | **0.00** | **0.00** | **0.03** | **0.00** | **0.00** | **0.03** | **0.00** | 0.98 | 0.07 | | 0.96 | **0.00** | 0.74 | 1.00 | **0.00** |
| 280 | 40 | **0.00** | **0.00** | 0.07 | **0.02** | 0.17 | 0.72 | **0.00** | **0.01** | 0.65 | **0.00** | 1.00 | 0.88 | 0.96 | | **0.01** | 1.00 | 1.00 | 0.23 |
| 280 | 60 | 1.00 | 1.00 | 1.00 | 1.00 | 1.00 | 0.69 | 0.96 | 1.00 | 0.75 | 1.00 | **0.00** | 0.49 | **0.00** | **0.01** | | **0.03** | **0.00** | 0.99 |
| 300 | 20 | **0.00** | **0.00** | 0.22 | 0.09 | 0.43 | 0.95 | **0.00** | **0.03** | 0.92 | **0.00** | 1.00 | 0.99 | 0.74 | 1.00 | **0.03** | | 1.00 | 0.52 |
| 300 | 40 | **0.00** | **0.00** | **0.01** | **0.00** | **0.03** | 0.25 | **0.00** | **0.00** | 0.21 | **0.00** | 1.00 | 0.42 | 1.00 | 1.00 | **0.00** | 1.00 | | **0.04** |
| 300 | 60 | 0.56 | 0.49 | 1.00 | 1.00 | 1.00 | 1.00 | 0.19 | 0.99 | 1.00 | 0.58 | 0.18 | 1.00 | **0.00** | 0.23 | 0.99 | 0.52 | 0.04 | |

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
