# Peer review of "Waste to Energy: Solid Fuel Production from Biogas Plant Digestate and Sewage Sludge by Torrefaction-Process Kinetics, Fuel Properties, and Energy Balance"

_energies, doi:10.3390/en13123161_

Round 1
Reviewer 1 Report
I enjoyed reading the manuscript entitled “Waste to Energy: Solid Fuel Production from Biogas Plant Digestate and Sewage Sludge by Torrefaction – Process Kinetics, Fuel Properties, and Energy Balance.” The study was well planned and executed. The collected data were properly analyzed and reported. The manuscript was well written. I suggest acceptance of the manuscript, with the following minor comments.
1st Comment: In Abstract, the authors used both “SS & D” (lines 22, 35, 35, and 37) and “D & SS” (lines 25, 27, and 29) to represent “sewage sludge and digestate” (or digestate and sewage sludge). I suggest the authors to use the short form more consistently, such as using “SS & D” throughout the manuscript.
2nd Comment: In Abstract, the authors mentioned clearly on lines 25-26 that they produced carbonized solid fuel (CSF) using SS or D at 200~300oC (interval 20oC), for 20~60 min (interval 20 min). However, they reported on lines 32-33 that “the terrified D had the highest HHV of 20 MJ.kg-1 under 300oC & 30 min...” It confused readers because the CSF was produced at 20, 40, or 60 min at different temperatures (but 30 min was not used as one of the testing conditions). I understand that the authors might interpolate the highest HHV to be achieved at 30 min from Figure 10 for D (i.e. digestate). Hence, I suggest the authors either to change “30 min” to “40 min” (see Table 2) or to clarify that “30 min” was “the interpolated value from the measured time periods”. “30 min” also appeared on line 534 (page 20) and line 621 (page 22).
3rd Comment: When the authors presented how energy densification ratio (EDr) was calculated, they used J.g-1 for HHVb but MJ.kg-1 for HHVa. Since they (J.g-1 and MJ.kg-1) are different by a factor of 1000, I suggest that the authors should either use J.g-1 or MJ.kg-1 for both HHVb and HHVa.
4th Comment: There are some other minor issues and typing mistakes.
- Line 54, Page 2: Please define the abbreviation “d.m.” (as dry mass) when it first appeared in text.
- Lines 78-79, Page 2: The sentence was incomplete […the shift from incineration and high-energy input (to what?)].
- Line 94, Page 2: The sentence should be rewritten because the phrase […and small plastics particles due so...] does not make any sense.
- Line 101, Page 3: “…some Ds from municipal biogas plants does…” should be written as “…some Ds from municipal biogas plants do…”
- Line 114, Page 3: “…D and SS…” should be written as “…SS and D…” (consistent with how the authors presented on line 116 (…SS and D…).
- Line 124, Page 3: See whether the authors can use “SS and D” or “SS & D” instead of “D&SS”.
- Line 264, Page 7: “regression coefficient” for (a2-a7) should be written as “regression coefficients”.
- Line 267, Page 7: “…each regression coefficients…” should be written as “…each regression coefficient…”
- Line 335, Page 9: “…up to 30 min…” please check whether it should be “…up to 40 min…” because the time intervals were set to 20 min, 40 min, etc.
- Figure 5: Remove ‘%’ from the vertical axis because it is a ratio.
- Line 363: Page 10: “Table S8” should be “Table A8”.
- Lines 420-421, Pages 12-13: The sentence “The k for 200~280oC range was higher for SS (k = 8.71.10-6 ~ 2.99.10-1)” was incorrect. Table 1 shows that k for 200oC for SS was 4.73.10-6 while k for 200oC for D was 4.91.10-6. So, the sentence should be “The k for 220~280oC was higher for SS…”
- Line 542, Page 20: “…A = 0.748 s-1 & A = 1.94 s-1…” should be written as “…A = 0.75 s-1 & A = 1.95 s-1…” (see Line 422 on Page 13 and Table 1).
- Line 609, Page 22: “…270 85 J.g-1…” should be “…27 085 J.g-1…”
- Line 615, Page 22: Please delete the phrase “but it should meet the standard it the case of these materials” because you did not measure the content of chlorine and mercury in your study. (An unsubstantiated claim should not be used in an academic article.)
- Line 726, References: “…EuropeaParliamentnt…” should be “…European Parliament…”
- Line 748, References: Delete the word “Environment” in front of “Removal”. The article title was “Removal of Antibiotics…”
- Line 752, References: Please provide the name and other details of ‘Conference’.
- Line 810, References: The title of this reference was incomplete. It should be “The Influence of Torrefaction Temperature on Hydrophobic Properties of Waste Biomass from Food Processing.”
- Line 821, References: The name of the first author should be “Pan, Z” (not “Zhongli, P”).
Reviewer 2 Report
Summarize the content of the manuscript
The manuscript reports the results of low temperature thermal conversion (torrefaction) of dry anaerobic digestate (D) and sewage sludge (SS) resulting in carbonized solid fuel (CSF).
The result indicated that the torrefied digestate had the highest HHV = 20 MJ∙kg-1 under 300 °C and 30 min, (HHV = 18 MJ∙kg-1 for unprocessed digestate). The torrefied sewage sludge has the highest HHV = 14.8 MJ∙kg-1 under 200 °C and 20 min (HHV 14.6 MJ∙kg-1 for unprocessed sewage sludge). The authors stated that the negative effect of torrefaction is an increase in ash content in CSF. This research has shown that there is a potential in using digestate as a substrate for torrefaction, whereas potential in using SS for fuel is questionable due to lack of HHV increase.
The manuscript’s strengths
The overal merit of the manuscript is very high. The paper is appropriate detailed and logically structured. The main chapters are good devided and aid the clarity of the text.
The tables and figures are easy to interpret, and support well the obtained results.
Comments
The production of fuels from biomass requires an energy input. The authors made an energy balance, but omitted an important stage in the preparation of the D and SS for torrefaction - drying and grinding. The digestate may contain only 3% dry matter. Drying this product will require a lot of energy, which may disqualify this product as a substrate for torrefaction. In future studies, authors should calculate the energy balance for the entire product / process life cycle.
Reviewer 3 Report
The article is addressing important and actual problem: waste-to-energy approaches and from this perspective different types of wastes and their transformation is important. Article is based on significant amount of experimental work and calculations and thus its publication would be important. Besides of minor mistakes (see attached file) there are some conceptual questions to be answered:
- The energy balance and all arguments are based on calculations considering use of dry (105 C) waste material. Is it really so??? If so, from practical use of these results this is a weakness as the water content in the sludge could be 90 - 95 %, compost ~ 60-70 % and from real application perspective the drying will produce a lot of water vapor and require huge energy. This aspect should be discussed in the text
- in the article should explained how the analysis and energetic value of torrefication gas has been done.
Otherwise the article is good and after answers on mentioned questions could be published
